# Characterization of cephalic and non-cephalic sensory cell types provides insight into joint photo- and mechanoreceptor evolution

**Roger Revilla-i-Domingo[1,2,3], Vinoth Babu Veedin Rajan[1,2], Monika Waldherr[1,2], Günther Prohaczka[1,2], Hugo Musset[1,2], Lukas Orel[1,2], Elliot Gerrard[4], Moritz Smolka[1,2,5], Alexander Stockinger[1,2,3], Matthias Farlik[6,7], Robert J Lucas[4], Florian Raible[1,2,3]\*, Kristin Tessmar-Raible[1,2]\***

[1]Max Perutz Labs, University of Vienna, Vienna BioCenter, Vienna, Austria; [2]Research Platform "Rhythms of Life", University of Vienna, Vienna BioCenter, Vienna, Austria; [3]Research Platform "Single-Cell Regulation of Stem Cells", University of Vienna, Vienna BioCenter, Vienna, Austria; [4]Division of Neuroscience & Experimental Psychology, University of Manchester, Manchester, United Kingdom; [5]Center for Integrative Bioinformatics Vienna, Max Perutz Labs, University of Vienna and Medical University of Vienna, Vienna, Austria; [6]CeMM Research Center for Molecular Medicine of the Austrian Academy of Sciences, Vienna, Austria; [7]Department of Dermatology, Medical University of Vienna, Vienna, Austria

**\*For correspondence:**
florian.raible@mfpl.ac.at (FR);
kristin.tessmar@mfpl.ac.at (KT-R)

**Abstract** Rhabdomeric opsins (r-opsins) are light sensors in cephalic eye photoreceptors, but also function in additional sensory organs. This has prompted questions on the evolutionary relationship of these cell types, and if ancient r-opsins were non-photosensory. A molecular profiling approach in the marine bristleworm *Platynereis dumerilii* revealed shared and distinct features of cephalic and non-cephalic *r-opsin1*-expressing cells. Non-cephalic cells possess a full set of phototransduction components, but also a mechanosensory signature. Prompted by the latter, we investigated *Platynereis* putative mechanotransducer and found that *nompc* and *pkd2.1* co-expressed with *r-opsin1* in TRE cells by HCR RNA-FISH. To further assess the role of r-Opsin1 in these cells, we studied its signaling properties and unraveled that r-Opsin1 is a Gαq-coupled blue light receptor. Profiling of cells from *r-opsin1* mutants versus wild-types, and a comparison under different light conditions reveals that in the non-cephalic cells light – mediated by r-Opsin1 – adjusts the expression level of a calcium transporter relevant for auditory mechanosensation in vertebrates. We establish a deep-learning-based quantitative behavioral analysis for animal trunk movements and identify a light– and r-Opsin-1–dependent fine-tuning of the worm's undulatory movements in headless trunks, which are known to require mechanosensory feedback. Our results provide new data on peripheral cell types of likely light sensory/mechanosensory nature. These results point towards a concept in which such a multisensory cell type evolved to allow for fine-tuning of mechanosensation by light. This implies that light-independent mechanosensory roles of r-opsins may have evolved secondarily.

## Introduction

Opsins, a subgroup of G protein-coupled transmembrane receptors (GPCRs), serve as the main light sensors in animal photoreceptor cells. Rhabdomeric opsins (r-opsins) are an ancient class of opsins

particularly widespread among invertebrates, typically expressed in larval photoreceptor cells and cephalic eyes that rely on rhabdomeric photoreceptors (*Arendt et al., 2002*; *Arendt and Wittbrodt, 2001*; *Ramirez et al., 2016*). The role of r-opsins in light perception has been best studied in the model of the *Drosophila* eye photoreceptor (EP) cells. Stimulation of a light-sensitive chromophore (retinaldehyde) covalently bound to the $G\alpha_q$-coupled r-opsin apoprotein initiates an intracellular cascade (with 12 key components of the phototransduction cascade) that leads to an increase in intracellular calcium (reviewed in *Hardie and Juusola, 2015*).

Whereas most of our knowledge about the function of r-opsins in animal photoreception stems from studies on cephalic EPs, non-cephalic *r-opsin*-expressing cells are found in representatives of various animal groups. For instance, *r-opsin* homologs demarcate putative photoreceptor cells at the tube feet of sea urchins (*Raible et al., 2006*; *Ullrich-Luter et al., 2011*), and in Joseph cells and photoreceptors of the dorsal ocelli of the basal chordate amphioxus (*Koyanagi et al., 2005*). In the case of the brittle star, such non-cephalic photoreceptor cells have been implicated in a form of vision (*Sumner-Rooney et al., 2020*). Yet the diverse locations of *r-opsin*-positive cells, and the fact that they are not strictly associated with pigment cells, have raised the question whether *r-opsin*-positive cells outside the eye might have different functional roles.

This implies an evolutionary question: to what extent do non-cephalic *r-opsin*-positive cells share an evolutionary history with cephalic EPs or represent independent evolutionary inventions? A biological context in which this question is particularly interesting to address are animals that exhibit segmented body axes, featuring sensory organs in some or all of these segments. Analyses of the early Cambrian Lobopodian fossil *Microdictyon sinicum* suggested that this putative ancestor of arthropods possessed compound eye structures above each pair of legs (*Dzik, 2003*; *Gehring, 2011*). This is in line with the idea that segmental photoreceptive organs could have been an ancestral feature, which might have been secondarily modified to allow for a more efficient division of labor between head and trunk. A similar hypothesis could be drawn for ancestors of annelids, a segmented clade of lophotrochozoans: various recent annelid groups, including opheliids, sabellids, and syllids, feature segmental eye spots with rhabdomeric photoreceptors (reviewed in *Purschke et al., 2006*). This would be consistent with the ancient presence of r-opsins and photoreceptive organs in a homonomously segmented annelid ancestor. Given the possible ancestry of segmentation in bilaterians (*Chen et al., 2019*; *Couso, 2009*; *Dray et al., 2010*), the outlined scenarios of segmental photoreceptive organs might even date back to the dawn of bilaterian animal evolution.

However, *r-opsin* genes have also started to be implied in functions that are unrelated to photoreception. Most notably, *r-opsin* genes are expressed in certain classes of mechanosensory cell types, such as the Johnston organ (JO) neurons and the larval Chordotonal organ (ChO) of *Drosophila* (*Senthilan et al., 2012*; *Zanini et al., 2018*), or the neuromasts of the lateral line of zebrafish and frog (*Backfisch et al., 2013*; *Baker et al., 2015*). Experiments assessing functionality of mechanosensation in both JO and ChO neurons have revealed that several *r-opsins* expressed in these receptors are required for proper mechanosensation and suggest that this function is light-independent (*Senthilan et al., 2012*; *Zanini et al., 2018*). These functional findings add new perspectives to older observations that a subset of mechanosensory cells (to which JO and ChO cells belong) exhibit significant similarities in their molecular specification cascade with EPs, comprising analogous use of Pax, Atonal, or Pou4f3 transcription factors (*Fritzsch, 2005*). If r-opsins are to be considered as part of a shared molecular signature in photosensory and mechanosensory cells, this raises divergent possibilities for the evolution of r-Opsin-positive sensory cells: (i) Could r-opsins have evolved as ancient 'protosensory' molecules that were primarily engaged in mechanosensation, only to secondarily evolve to become light receptive? (ii) Conversely, does the canonical function of r-opsins in light reception reflect their ancestral role, with r-opsin-dependent mechanoreception representing a secondary evolutionary modification? Or (iii) are there ways in which photosensory functions of r-opsins could have played an ancient role in mechanoreception, even if this role might not be present any more in the investigated *Drosophila* mechanoreceptors?

In order to gain insight into these questions of r-opsin function, and into the evolution of sensory systems from an independent branch of animal evolution, we characterized *r-opsin*-expressing cells in a lophotrochozoan model system, the marine annelid *Platynereis dumerilii* that is amenable to functional genetic analyses (*Bannister et al., 2014*; *Bezares-Calderón et al., 2018*; *Gühmann et al., 2015*). After its pelagic larval stage, *P. dumerilii* inhabits benthic zones (*Fischer and Dorresteijn,*

*2004*). These are characterized by a complex light environment, making it likely that light sensory systems have been evolutionarily preserved in this model, rather than being secondarily reduced. In line with this, *Platynereis* has retained an evolutionarily representative set of *r-opsins* (*Arendt et al., 2002*; *Randel et al., 2013*) and other photoreceptor genes. The *Platynereis r-opsin1* gene is not only expressed in EPs (*Arendt et al., 2002*), but also in peripheral cells along the trunk of the animal (*Backfisch et al., 2013*) (cells referred to in this study as trunk *r-opsin1* expressing [TRE] cells), making the worm an attractive species for a comparative assessment of r-opsin function between cephalic and non-cephalic cell types. While both EP and TRE cells express the *gaq* gene that encodes a G$_{\alpha q}$ subunit (*Backfisch et al., 2013*), it has remained elusive whether the TRE cells represent a segmental repetition of the EP cell type along the body plan or represent a distinct sensory modality. Likewise, it is unclear whether r-Opsin1 in TRE cells has a light sensory role, as in the EPs, or serves a light-independent function, as has been suggested for *Drosophila* JO or ChO neurons.

Here, we established a dissociation and fluorescence-activated cell sorting (FACS) protocol for the *Platynereis* pMos{rops::egfp}$^{vbci2}$ strain that expresses enhanced GFP (EGFP) under the regulatory control of the *Platynereis r-opsin1* gene in both EP and TRE cells (*Backfisch et al., 2013*). Molecular profiling of both EP and TRE cells revealed that TRE cells, but not EP cells, possess a mechanosensory signature. Building on a novel hybridization chain reaction RNA fluorescent in situ hybridization (HCR RNA-FISH) approach (*Choi et al., 2018*; *Kuehn et al., 2021*), we confirm expression of the mechanical transducing factor orthologs *nompc* and *pkd2.1* in TRE cells. Targeted mutagenesis of the endogenous *r-opsin1* locus and an experimental characterization of the r-Opsin1 action spectrum allowed us to uncover that, specifically in the TRE cells, light – mediated by r-Opsin1 – adjusts the expression level of a plasma membrane calcium transporter relevant for auditory mechanosensation in vertebrates. Our data, therefore, suggest that TRE cells represent a distinct mechanoreceptive cell type, in which r-Opsin1, in difference to the current *Drosophila*-based paradigms, elicits light-dependent functional changes. In line with this, a newly established deep-learning-based approach revealed light-dependent behavioral differences between wildtype and *r-opsin1* mutant trunks. Our results are consistent with the idea that photo- and mechanosensory systems have a common evolutionary origin in a multimodal sensory cell type.

## Results

### Shared and distinct molecular signatures of EP and TRE cells

In order to gain insights into the molecular signatures of EP and TRE cells, we established a mechanical dissociation protocol compatible with FACS and benchmarked to minimize cell death. We next dissected heads and trunks of the same pMos{rops::egfp}$^{vbci2}$ individuals (*Figure 1A*), isolated EGFP-positive cells from heads and trunks, and established transcriptomes for both sorted and unsorted cells using Illumina HiSeq sequencing on cDNA amplified by the Smart-Seq2 protocol (*Picelli et al., 2014*; *Figure 1B*). Gates for FACS (*Figure 1C, D*) were calibrated using dissociated cells from wildtype heads (*Figure 1—figure supplement 1A*) and trunks (*Figure 1—figure supplement 1B*) to exclude isolation of autofluorescent cells.

To validate the sampling strategy, we investigated if this procedure reproduced expected results for genes known to be enriched in both EP and TRE cells. Both *r-opsin1* and *egfp* were up to several thousand times more abundant in libraries derived from EGFP-positive cells than in those of unsorted cells (*Figure 1E, F*). In further support of successful enrichment, signatures of EGFP-positive cells were consistently enriched in the *gq* gene encoding the G alpha subunit *Gαq* (*Figure 1G*). *Gq* was previously shown to be strongly expressed in EP and TRE cells (*Backfisch et al., 2013*). By contrast, the genes encoding the ribosomal subunit Rps9 or the Polo-like Kinase Cdc5, previously established as internal controls for gene expression quantification experiments (*Zantke et al., 2013*), were not enriched in either EP or TRE cell populations (*Figure 1—figure supplement 1C, D*).

As these results indicated that the experimental procedure allowed for significant enrichment and profiling of EP and TRE cells, we next used EdgeR (*Robinson et al., 2010*) to systematically calculate enrichment scores for each of the EGFP-positive populations compared to the combined set of head and trunk unsorted cells. From a total of 39,575 genes, we determined a set of 278 genes (0.7%) to be significantly enriched in EP cells and a set of 361 genes (0.9%) significantly enriched in TRE cells (False Discovery Rate [FDR] < 0.05) (*Figure 2A*). 133 genes (0.3% of total) were shared between the

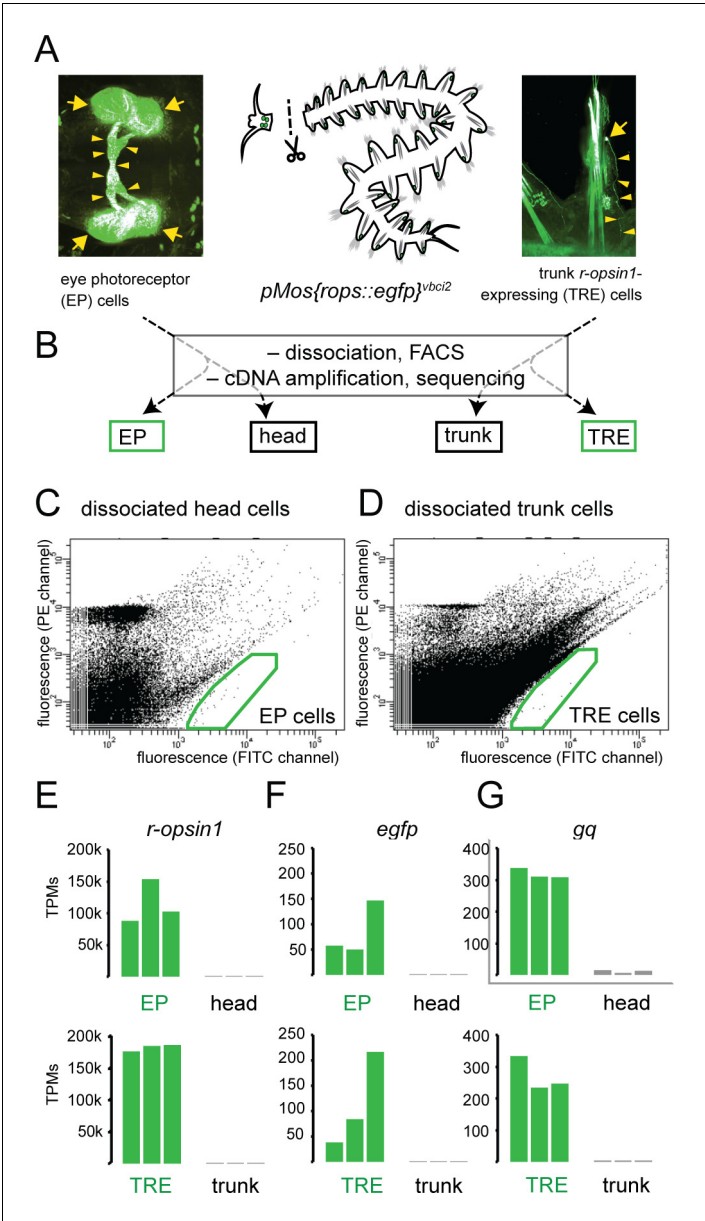

**Figure 1.** Establishment of molecular signatures of eye photoreceptors (EPs) and trunk *r-opsin1*-expressing (TRE) cells. (**A**) Dissection of pMos{rops::egfp}^vbci2 individuals, separating the head containing EP cells (left panel) from trunk containing TRE cells (right panel). (**B**) Overview of cDNA library generations, resulting in fluorescence-activated cell sorting (FACS)-enriched (EP, TRE) and unsorted (head, trunk) samples. (**C, D**) Representative FACS plots showing gated populations (green boxes) of EP and TRE cells, respectively. For non-transgenic controls, see *Figure 1—figure supplement 1A, B*. (**E–G**) Comparison of transcripts per million reads (TPM) for the genes *r-opsin1* (**E**), *enhanced green fluorescent protein/egfp* (**F**), and *gαq/gq* (**G**) in individual replicates of EP, head, TRE, and trunk libraries. For comparison of TPMs for non-enriched control genes, see *Figure 1—figure supplement 1C, D*. Arrows and arrowheads in (**A**) designate EGFP-positive cell bodies and projections, respectively.

The online version of this article includes the following figure supplement(s) for figure 1:

**Figure supplement 1.** Fluorescence-activated cell sorting (FACS) profiles of dissociated cells from wild-type heads and trunks.

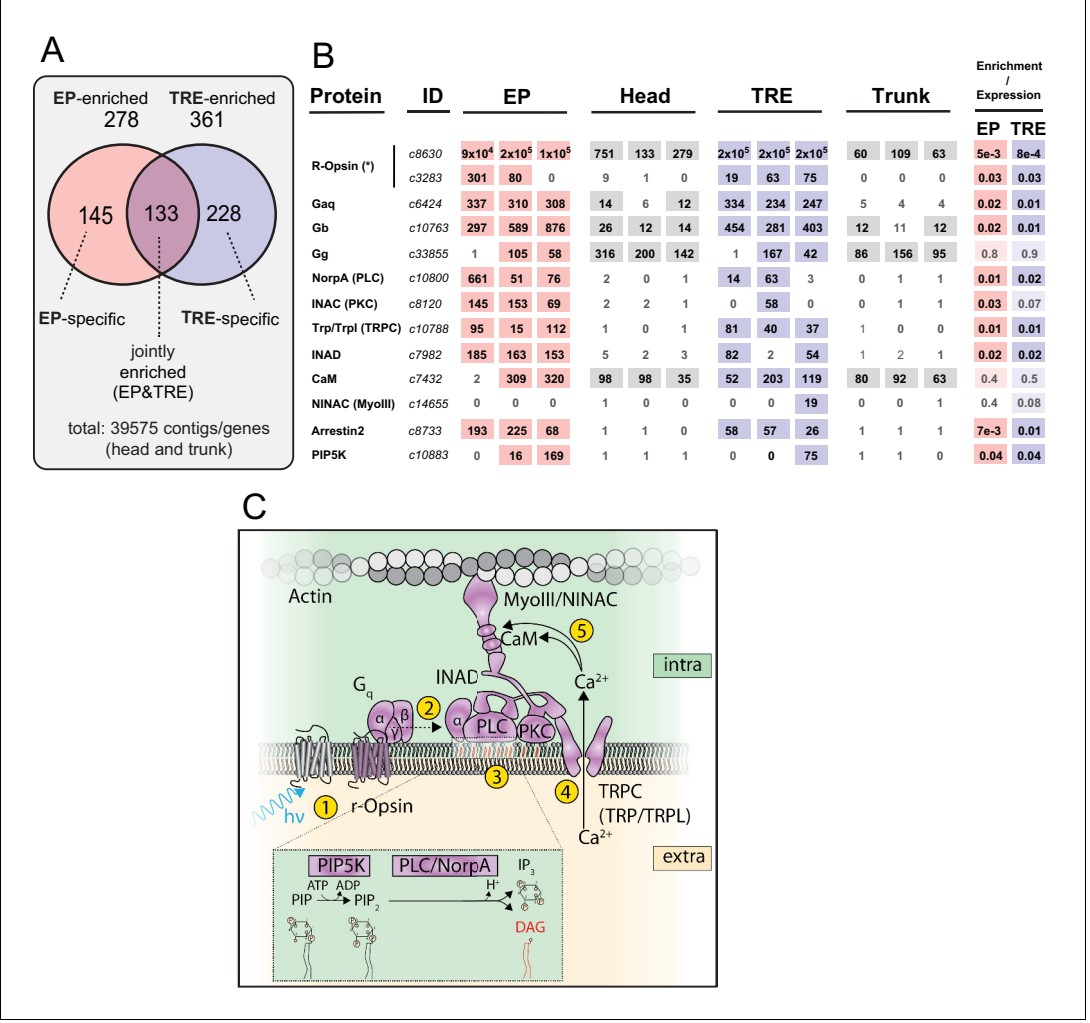

**Figure 2.** Trunk *r-opsin1*-expressing (TRE) cells share critical elements of the phototransduction cascade with eye photoreceptors (EPs). (**A**) Summary of gene/contig counts enriched in EP and TRE cells compared to the combined background (head and trunk; 39,575 contigs) and their respective overlap. For expression levels of genes selected for validation, see *Figure 2—figure supplement 1*. For validation experiments, see 'Analysis and validation of differentially expressed genes,' *Figure 2—figure supplements 2* and *3*, *Supplementary file 1*. (**B**) Expression and enrichment of phototransduction components in EP and TRE cells. Protein: key components of the *D. melanogaster* phototransduction pathway (cf. panel **C**). ID: corresponding gene ID(s) of the *P. dumerilii* transcriptome. EP, head, TRE, trunk: expression levels (in transcripts per million reads [TPMs]) of the respective gene in individual replicates of *P. dumerilii* EP, head, TRE, or trunk libraries, respectively. Light shades indicate expression above the established threshold. Enrichment/expression: FDR values obtained from the differential expression analysis for EP or TRE cells. Dark shades indicate significant enrichment, light shades expression without significant enrichment in the respective cell type. IDs *c8630* and *c3283* (demarcated with asterisk) relate to two distinct *r-opsin* genes expressed by EP and TRE cells: *r-opsin1* (AJ316544.1) and *r-opsin3* (KC810971.1), respectively. Note that although the *P. dumerilii* best BLAST hit to NINAC/MyoIII (*c14655*, E value: 3e-166) is not expressed in EP cells, the second best BLAST hit (*c8565*, E value: 1e-64) is expressed in these cells. For sequence identifiers of the relevant *P. dumerilii* genes, see *Supplementary file 2*. (**C**) Scheme highlighting factors present or enriched in the joint EP/TRE signature (cf. panel **B**), and their function in critical steps (yellow circles 1–5) of the canonical r-opsin phototransduction cascade. Enlarged inset shows relevant enzymatic steps in the intracellular leaflet. (**C**) Modeled after *Hardie and Juusola, 2015*.

The online version of this article includes the following figure supplement(s) for figure 2:

**Figure supplement 1.** Expression levels (in transcripts per million reads [TPMs], in individual replicates) of enriched genes chosen for validation.

**Figure supplement 2.** Validation of selected genes from the differential enrichment analysis (head).

**Figure supplement 3.** Validation of selected genes of the differential enrichment analysis (trunk).

EP and TRE cells (common EP-/TRE-enriched genes), including, expectedly, *r-opsin1* and *gq* (*Supplementary file 1*), and leaving 145 (0.4% of total) EP-specific genes and 228 (0.6% of total) TRE-specific genes (*Figure 2A*). Experiments on selected genes (see 'Analysis and validation of differentially expressed genes,' *Figure 2—figure supplement 1*, *Supplementary file 1*) allowed us to validate the specificity of the predicted sets (*Figure 2—figure supplement 2*, *Figure 2—figure supplement 3*), pointing at both shared and distinct properties of the *r-opsin1*-expressing cells of the head and the trunk.

Previous analyses had already revealed several genes to be expressed in adult EPs: *vesicular glutamate transporter* (*vglut*) (*Randel et al., 2014*; *Tomer et al., 2010*), the rhabdomeric opsin gene *r-opsin3* (*Randel et al., 2013*), the G$_o$-type opsin gene *G$_o$-opsin1* (*Ayers et al., 2018*; *Gühmann et al., 2015*), and the light-receptive cryptochrome gene *l-cry* (*Zantke et al., 2013*). Sequences of *r-opsin3*, *G$_o$-opsin1*, and *l-cry* were all significantly enriched in the EP-derived transcriptome when compared to unsorted head cells. While *r-opsin3* has already been described to be expressed in TREs (*Randel et al., 2013*), also *G$_o$-opsin1*and *l-cry* were part of the specific TRE transcriptome, pointing at an unexpected complexity of light receptors in the TRE cells, and a similar equipment of EP and TRE cells with photoreceptive molecules. Our sequencing data did not cover the published *vglut* gene in any of the samples (possibly reflecting low expression in the adult).

As to potential differences between EPs and TRE cells, prior analyses had pointed to the expression of circadian clock genes in the EPs and the adjacent brain lobes (*Zantke et al., 2013*), and both classical and molecular studies suggested the retina as a site of continuous neurogenic activity (*Fischer and Brökelmann, 1966*; *Pende et al., 2020*), contrasting with the appearance of the TRE cells as sparse, differentiated neurons (*Backfisch et al., 2013*). In line with these expectations, we found the EP-, but not TRE-derived transcriptomes to be enriched, respectively, in the circadian clock gene *bmal*, as well as a homolog of the *embryonic lethal, abnormal vision/elav* gene, a marker characteristic for committed neurons (*Denes et al., 2007*; *Kerner et al., 2009*).

## EP and TRE cells share a complete phototransduction pathway

Building on these initial results, we next explored if the identified gene sets could provide additional insights into the function and evolution of the TRE cells. We first assessed whether molecular data in addition to the identified photoreceptor molecules would support a possible function of TRE cells in light sensitivity as it would be expected if these cells represented segmentally repeated cell-type homologs of the *P. dumerilii* and *Drosophila melanogaster* EP cells. To test this hypothesis, we compared EP- and TRE-enriched genes of our bristleworm with a published set of genes enriched in *Drosophila* EP cells (*Yang et al., 2005*).

Using BLAST-based homology relationships between *D. melanogaster* and *P. dumerilii* genes (see Materials and methods), we established a set of 408 bona fide *P. dumerilii* homologs of the 743 *D. melanogaster* EP-enriched genes. Nine of these were common EP-/TRE-enriched genes. A statistical analysis, based on the generation of $10^4$ sets of 743 randomly picked *D. melanogaster* genes (see Materials and methods), indicated that this number of common EP-/TRE-enriched genes significantly exceeds random expectation (*Figure 3—figure supplement 1*, p=0.024). Among these nine overlapping genes, we found five bona fide *P. dumerilii* homologs of genes considered key components of the r-opsin phototransduction pathway described for *Drosophila* EP cells (*Hardie and Juusola, 2015*) (yellow box in *Figure 3—source data 1*). By extending our assessment to bona fide homologs of additional components of the *Drosophila* r-opsin phototransduction pathway, we found that putative homologs of 9 and 8 of the 12 key components of the r-opsin phototransduction pathway are enriched in the *P. dumerilii* EP and TRE cells, respectively (*Figure 2B*). Statistical analysis with $10^4$ random gene sets of matching size (see Materials and methods) revealed these results to be highly significant (p<$10^{-4}$, for both EP and TRE). Of note, all 12 key components of the r-opsin phototransduction pathway were found to be expressed in the TRE cells of *P. dumerilii* (*Figure 2B, C*; p<$10^{-4}$).

## TRE cells combine photo- and mechanosensory molecular signatures

Following the same strategy, to further explore potential additional functions of the TRE cells, we next tested the molecular relationship between the worm's TREs and the r-opsin-expressing, mechanosensory JO neurons of *Drosophila*. For this, we took advantage of 101 genes identified as JO neuron specific in a microarray analysis (*Senthilan et al., 2012*) and 80 *P. dumerilii* homologs of these.

Significant subsets of these were found in the common EP-/TRE-enriched signature (nine genes; p<10$^{-4}$), and the TRE-specific signature (seven genes; p=1.3 $\times$ 10$^{-3}$) (*Figure 3A*). The common EP-/TRE-enriched genes essentially reflect the *P. dumerilii* homologs of the aforementioned phototransduction pathway (*rh3/rh4, rh5/rh6, trp/trpl, norpa, gβ76c, pip5k59b, arr2,* and *klp68D; Figure 3B*). This finding underlines the similarity of TRE cells with JO neurons. notably on the level of the phototransduction machinery.

Given the well-established function of JO neurons as mechanosensory cells, we next investigated whether the additional, statistically significant overlap between JO-specific genes and TRE-specific genes reflected any shared mechanosensory signature. Among the seven JO-specific genes overlapping with TRE-specific genes, two were shown (*Senthilan et al., 2012*) to be required for the normal function of JO neurons (*gl* and *wtrw*; matching *Platynereis c2053* and *trpA/c7677*, respectively; *Figure 3B–E*), adding to the four (of nine) specifically shared genes from the joined TRE/EP set with known mechanical functions (asterisks in *Figure 3B*), whereas the other five have not been tested for mechanosensory functions. In order to compensate for this lack of functional information, we also performed a comparison with mouse, where the largest number of genes involved in mechanosensation is known. We systematically determined putative *P. dumerilii* homologs of all mouse genes assigned to the gene ontology (GO) category 'sensory perception of mechanical stimulus' (GO:0050954), and then assessed their overlap with EP- or TRE-expressed genes (*Figure 3F, G*). Indeed, these bona fide homologs are significantly overrepresented in the TRE-specific signature (p=0.029; *Figure 3G*, for gene IDs, see *Figure 3—source data 2*). Similar analyses with GO categories for other sensory perception modalities associated to the JO, such as 'sensory perception of temperature stimulus' (GO:0050951) and 'sensory perception of pain' (GO:0019233), showed no statistically significant results (*Figure 3H, I*).

A closer analysis of those mouse mechanosensory genes whose bristleworm counterparts are expressed in TRE cells (*Figure 3—source data 2*) points at a gene signature shared between TRE cells and mouse inner ear hair (IEH) cells: 18 out of the 19 TRE-specific gene homologs have reported effects on hearing function in the mouse (yellow shading in *Figure 3—source data 2*). Notably, *whrn, dnm1*, *atp8b1, myo3a, chrna9,* and *tecta* have been shown to be required for vertebrate IEH cell function (*Boëda et al., 2002*; *Neef et al., 2014*; *Schneider et al., 2006*; *Stapelbroek et al., 2009*; *Zou et al., 2014*); *sox2* and *jag2* are known to be required for development of vertebrate IEH cells (*Gou et al., 2018*; *Kiernan et al., 2005*); *crym, serpinb6a,* and *myh14* lead to hearing loss when mutated in mammals (*Donaudy et al., 2004*; *Fu et al., 2016*; *Oshima et al., 2006*; *Tan et al., 2013*). Again, we assessed the specificity of this finding by systematically comparing the overlap with mouse genes involved in distinct modalities of mechanosensation, confirming a statistically significant overlap between mouse hearing genes and *P. dumerilii* TRE-specific genes (p=0.0087; *Figure 3J*), whereas other mechanosensory modalities yielded no statistically significant results (*Figure 3K*). Even though additional functions, unrelated to mechanosensation, are known for some of the above genes, these statistical results strongly argue for a gene signature specifically shared between *P. dumerilii* TRE cells and mouse IEH cells.

The shared mechanosensory transcriptome signature of *P. dumerilii* TRE cells, *Drosophila* JO neurons, and mouse IEH cells is consistent with the possibility that TRE cells retain a combination of mechano- and photosensory molecular features as they were previously suggested to form a likely ancestral protosensory state (*Fritzsch et al., 2007*; *Fritzsch, 2005*; *Niwa et al., 2004*). The notion of a likely evolutionarily meaningful molecular relationship between these cells is further reinforced by the observation that the worm's TRE cells differentiate out of a territory that expresses the gene encoding for the transcription factor Pax2/5/8 (*Backfisch et al., 2013*). Differentiation of JO neurons and mouse IEH cells requires respective *Drosophila* (*spa*) and mouse (*pax2*) orthologs. Moreover, TRE cells have been linked to expression of *brn3/pou4f3* (*Backfisch et al., 2013*), a *Platynereis* ortholog of the vertebrate *pou4f3* gene. *Pou4f3* demarcates the neuromasts of the fish lateral line (*Xiao et al., 2005*), a set of mechanosensory structures that also express fish r-opsin orthologs (*Backfisch et al., 2013*).

## *P. dumerilii* TRE cells express mechanical transducing molecules

While we found statistically significant photo- and mechanosensory molecular signatures in the TRE cells, transcripts of putative or validated mechanical stimulus transducing molecules were not among the identified molecules. We reasoned that like in other mechanosensory cells, expression levels of

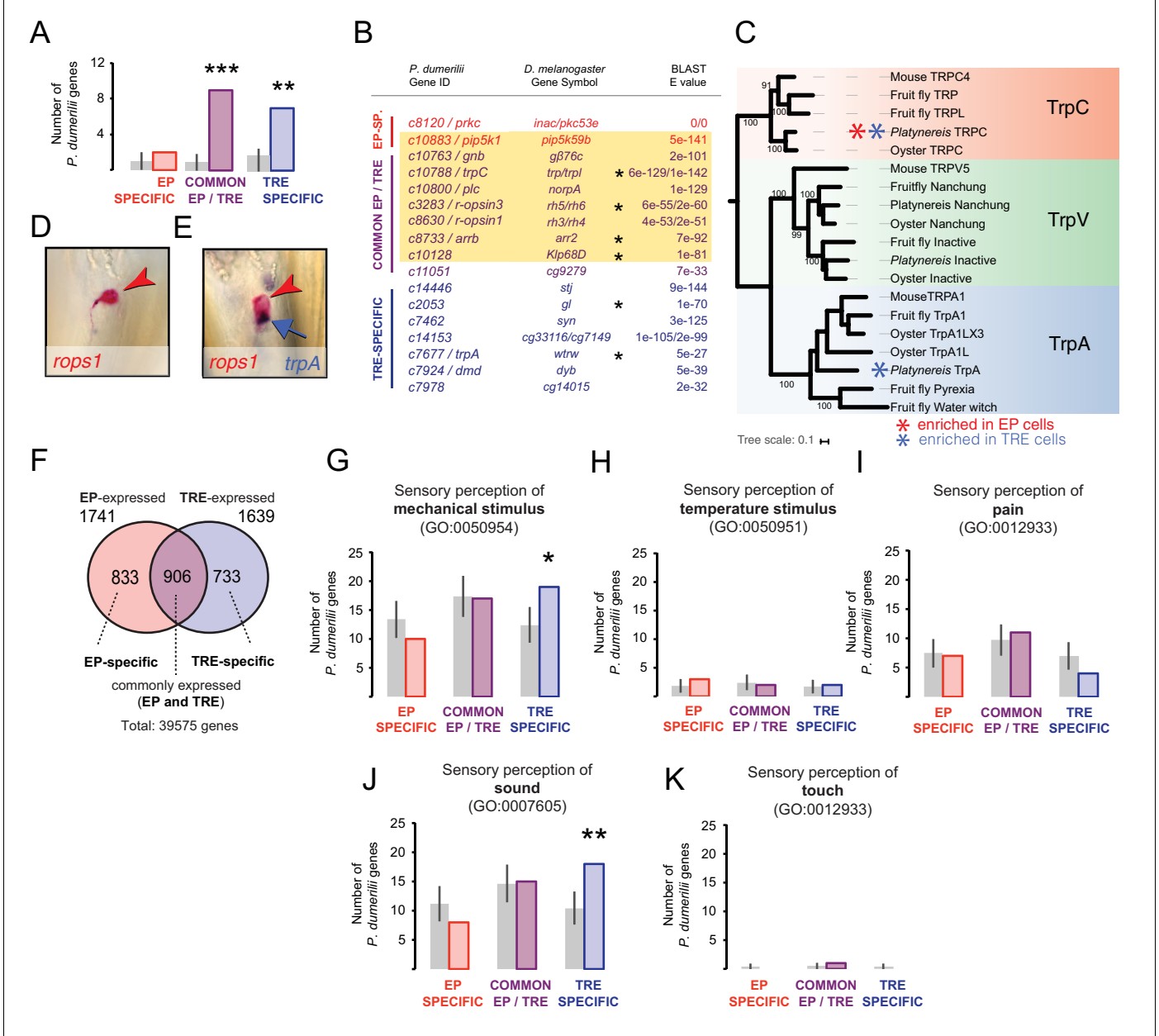

**Figure 3.** Trunk *r-opsin1*-expressing (TRE) cells also share a mechanosensory signature. (A, B) Comparison of *P. dumerilii* eye photoreceptor (EP)- and TRE-enriched genes with *D. melanogaster* Johnston organ (JO)-enriched genes. For comparison with *D. melanogaster* EP-enriched genes, see *Figure 3—figure supplement 1*, *Figure 3—source data 1*. (A) Number of *P. dumerilii* genes specifically enriched in EP cells (EP-specific, red), enriched in EP and TRE cells (common EP- and TRE-enriched, purple), or specifically enriched in TRE cells (TRE-specific, blue) overlapping with *D. melanogaster* JO-enriched genes. Gray bars show the average number (± standard deviation) of TRE-specific, common EP- and TRE-enriched or TRE-specific *P. dumerilii* genes overlapping with randomly selected sets of *D. melanogaster* genes. **p<0.01; ***p<10⁻⁴. (B) List of the overlapping genes indicated in (A). Each gene in the '*P. dumerilii* Gene ID' column indicates the best *P. dumerilii* BLAST hit of the listed *D. melanogaster* gene. The yellow shading indicates genes that are part of the *D. melanogaster* phototransduction pathway. Asterisks indicate genes relevant for auditory JO function (*Senthilan et al., 2012*). (C) Molecular phylogeny of Transient receptor potential channel (Trp) orthologs showing the assignment of the joint EP/TRE-enriched TRPC channel, and the *Platynereis* TrpA ortholog expressed in the TRE cells. For sequence identifiers see *Supplementary file 3*. (D, E) Specific co-expression of *Platynereis r-opsin1* (D, E, red; red arrowheads) and *trpA* (E, purple; blue arrow) in TRE cells, reflecting one of various TRE markers shared with mechanosensory cells (see B); caudal views, distal to the top. (F) Number of genes expressed in EP and/or TRE cells. (G–K) Number of EP-specific (red), common EP-/TRE-expressed (purple), or TRE-specific (blue) *P. dumerilii* genes overlapping with *Mus musculus* genes involved in sensory perception of mechanical stimulus (G), sensory perception of temperature stimulus (H), sensory perception of pain (I), sensory perception of sound (J), or sensory perception of touch (K). For identity of overlapping genes indicated in (G), see *Figure 3—source data 2*. For list of overlapping genes indicated in (J), see *Figure 3—source data 2* (yellow shading). Gray bars show the average number (± standard deviation) of TRE-

*Figure 3 continued on next page*

Figure 3 continued

specific, common EP-/TRE-enriched, or TRE-specific *P. dumerilii* genes overlapping with randomly-selected sets of *M. musculus* genes. * p<0.05; ** p<0.01.

The online version of this article includes the following source data and figure supplement(s) for figure 3:

**Source data 1.** List of the overlapping genes identified in the comparison of *P. dumerilii* eye photoreceptor (EP-) and trunk *r-opsin1*-expressing (TRE)-enriched genes with *D. melanogaster* EP-enriched genes.

**Source data 2.** List of the overlapping genes indicated in *Figure 3G*.

**Figure supplement 1.** Comparison of *P. dumerilii* eye photoreceptor (EP-) and trunk *r-opsin1*-expressing (TRE)-enriched genes with *D. melanogaster* EP-enriched genes.

such molecules might be very low, particularly upon terminal differentiation (*Arnadóttir and Chalfie, 2010*). We thus assessed expression of selected mechanical transducing molecules in posterior regenerates, considering that expression levels might be higher in developing cells that rapidly reform.

We first identified, from available transcript sequences, *Platynereis* orthologs of the four main gene families previously shown to function as mechanical transducing molecules in animals (*Fritzsch et al., 2020*), omitting the class of MscS-like factors so far only implicated as animal mechanical transducing molecules in the cnidarian *Nematostella* (*Fritzsch et al., 2020*). Besides Pkd2.1 – previously shown to function as a mechanical transducing molecule in *Platynereis* larvae (*Bezares-Calderón et al., 2018*) – our analysis included *Platynereis* orthologs of the TRP-family member NOMPC (Pdu-NOMPC; *Figure 4—figure supplement 1A*), the PIEZO-family (Pdu-Piezo; *Figure 4—figure supplement 1B*), and of the TMC-family (Pdu-TMC1/2/3; *Figure 4—figure supplement 2*). Except for TMC1/2/3, the presence of the respective coding sequences in posterior regenerates was validated by cloning. To assess (co-) expression of the validated genes in the regenerate, we adapted the new technique of in situ hybridization chain reaction (*Choi et al., 2018*) that has already been successfully used on *Platynereis* germ cells (*Kuehn et al., 2021*). To benchmark the procedure for regenerates, we performed a side-by-side detection of *Platynereis r-opsin1* by conventional in situ hybridization and in situ HCR in regenerates. This confirmed the reliability of the technique (*Figure 4—figure supplement 3*). We next analyzed the staining of *Pdu-nompc, Pdu-piezo,* and *Pdu-pkd2.1* via in situ HCR (*Figure 4—figure supplement 4*). Whereas the negative control was devoid of staining (*Figure 4—figure supplement 4A, E*), all three genes showed cellular staining in the regenerate, with *Pdu-nompc* (*Figure 4—figure supplement 4B, F*) and *Pdu-pkd2.1* (*Figure 4—figure supplement 4D, H*) being much more restricted than *Pdu-piezo* (*Figure 4—figure supplement 4C, G*). *Pdu-piezo* showed an almost ubiquitous expression, with some cells exhibiting slightly stronger staining than others (*Figure 4—figure supplement 4C, G*). Although this gene therefore likely co-expressed with *Pdu-r-opsin1*, we reasoned that it did not add much to a specific cell-type signature, and hence did not proceed with a more detailed analysis. Instead, we focused on co-expression analyses of *r-opsin1* and *nompc*, as well as of *r-opsin1* and *pkd2.1*, respectively. The negative control for the HCR amplifiers showed no detectable staining (*Figure 4A–F*). By contrast, both *Pdu-nompc* (*Figure 4G–L*) and *Pdu-pkd2.1* (*Figure 4M–R*) co-expressed with *Pdu-r-opsin1* in individual TRE cells (arrowheads in *Figure 4G–I, M–O*). In both cases, we also detected cells that do not co-express *Pdu-r-opsin1*, indicative of a diversity of putative mechanosensory cells in *Platynereis* trunks. Given the functional validation of Pkd2.1 as a mechanoreceptive molecule in *Platynereis* larvae (*Bezares-Calderón et al., 2018*) and the functional validation of NOMPC in other species (*Fritzsch et al., 2020*; *Walker et al., 2000*), our data further support the notion that TRE cells are indeed functional mechanoreceptors. How the cells expressing different mechanical transducing molecules relate to each other remains open. The scenario in *Platynereis* might be similar to flies and nematodes, where both TRP and DEG/ENaC channels function in mechanosensation without a clear connection of either channel with specific receptor properties (*Fritzsch et al., 2020*).

## *P. dumerilii* r-Opsin1 mediates blue light reception through Gα$_q$ signaling

As a prerequisite for a deeper analysis of the function of r-Opsin1 in the TRE cells of *P. dumerilii*, we next set out to characterize the photosensory properties of this opsin. A distinctive feature of r-opsin phototransduction cascade is the coupling and light-dependent activation of the Gα$_q$ protein by

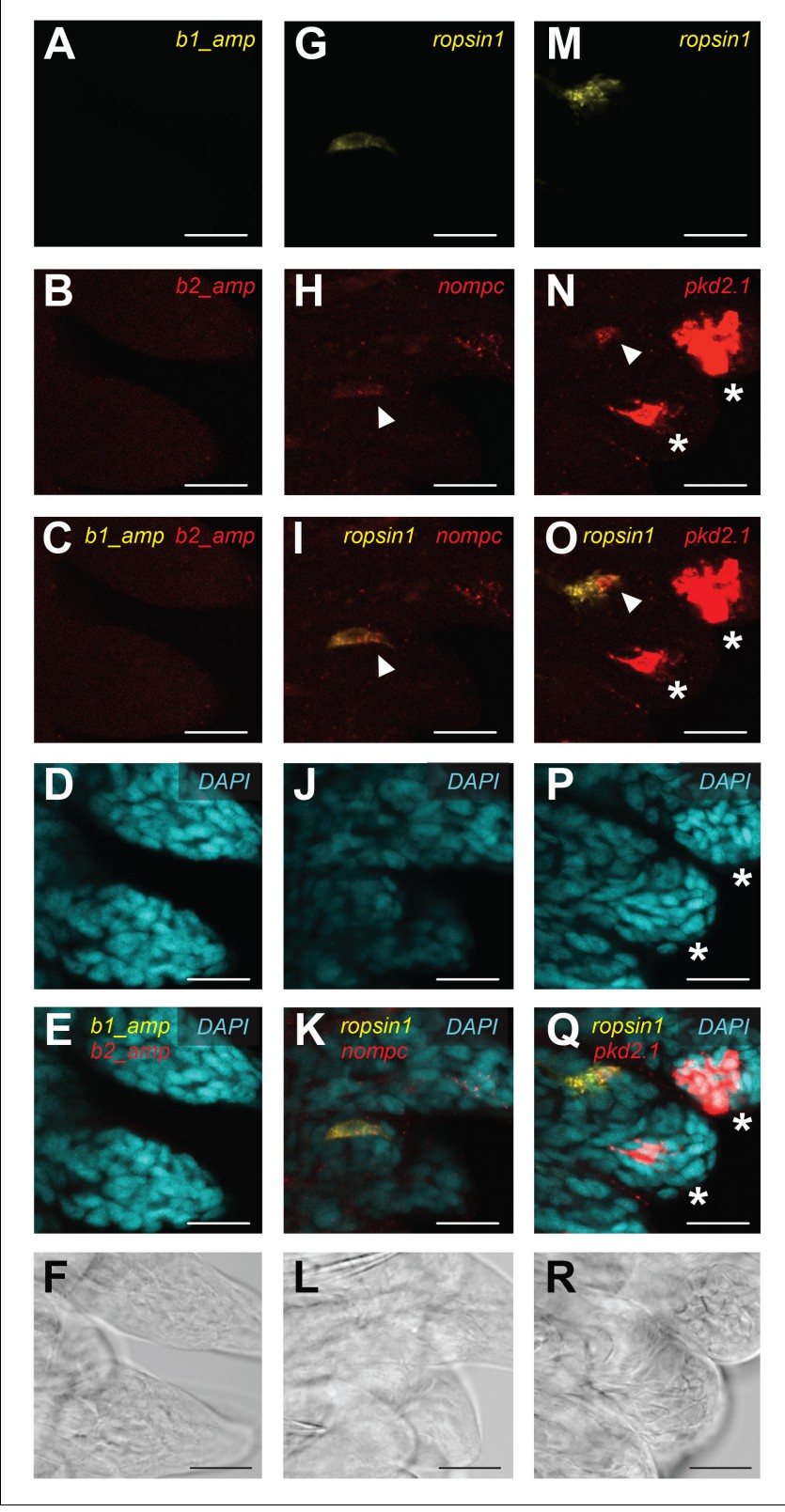

**Figure 4.** Expression of *nompc* and *pkd2.1* in trunk *r-opsin1*-expressing (TRE) cells of trunk regenerates. Hybridization chain reaction (HCR) confocal microscope images of *Platynereis* trunk regenerates showing background fluorescence by the fluorescently labeled HCR amplifier hairpins b1 and b2 (**A–F**) or the HCR fluorescence signal in samples exposed to a b1-coupled *r-opsin1* probe and a b2-coupled *nompc* probe (**G–L**) or

*Figure 4 continued on next page*

*Figure 4 continued*

a b1-coupled *r-opsin1* probe and a b2-coupled *pkd2.1* probe (M–R). (**A, G, M**) HCR signal (Alexa Fluor 546 fluorescence) from the b1 amplifier hairpins, not associated to any probe (**A**), or associated to a *r-opsin1* probe (**G, M**). (**B, H, N**) HCR signal (Alexa Fluor 647 fluorescence) from the b2 amplifier hairpins, not associated to any probe (**B**), or associated to a *nompc* probe (**H**) or a *pkd2.1* probe (**N**). The arrowheads in (**H**) and (**N**) indicate, respectively, the *nompc* and *pkd2.1* that overlap with the *r-opsin* signal. The asterisks in (**N**) indicate strong *pkd2.1* expression in the tips of the developing parapodia, consistent with the overview image shown in *Figure 4—figure supplement 3*. (**C, I, O**) Overlap of HCR signals from the b1 and b2 amplifier hairpins in (**A**) and (**B**), (**G**) and (**H**), or (**M**) and (**N**), respectively. Arrowheads in (**I**) and (**O**) indicate the same position as in (**H**) and (**N**), respectively. The asterisks in (**O**) indicate the same positions as in (**N**). (**D, J, P**) Fluorescence signal generated by the DAPI counterstain. The asterisks in (**P**) indicate the same positions as in (**N**) and (**O**). (**E, K, Q**) Overlap of HCR signals from the b1 amplifier hairpins, b2 amplifier hairpins, and DAPI counterstain in (**A, B, D**), (**G, H, J**), or (**M, N, P**), respectively. The asterisks in (**Q**) indicate the same positions as in (**N, O, P**). (**F, L, R**) Transmitted light (T-PMT) corresponding to the same field of view as in (**A–E**), (**G–K**), and (**M–Q**), respectively. Scale bars: 20 μm.

The online version of this article includes the following figure supplement(s) for figure 4:

**Figure supplement 1.** Molecular phylogeny identifies *Platynereis dumerilii* NompC and Piezo orthologs.

**Figure supplement 2.** Molecular phylogeny identifies a *Platynereis dumerilii* Tmc123 ortholog.

**Figure supplement 3.** Validation of the hybridization chain reaction (HCR) 3.0 technique in *Platynereis dumerilii* trunk regenerates.

**Figure supplement 4.** Overview expression pattern of *nompc, piezo,* and *pkd2.1* in *Platynereis* trunk regenerates.

r-opsin (*Scott et al., 1995*). Amphioxus, chicken, and human melanopsins – orthologs of *Drosophila* r-opsins – have all been shown to elicit intracellular calcium increase in response to light, and that all of these are capable of activating the G$\alpha_q$ protein in a light-dependent manner (*Bailes and Lucas, 2013*). Given that *P. dumerilii* r-Opsin1 is an ortholog of *Drosophila* r-opsins and chordate melanopsins (*Arendt et al., 2004*; *Arendt et al., 2002*), we tested if *P. dumerilii* r-Opsin1 can also activate G$\alpha_q$ signaling upon light exposure by employing a cell culture second messenger assay (*Bailes and Lucas, 2013*) (see Materials and methods). *P. dumerilii r-opsin1*-transfected HEK293 cells exhibited a significant response to light exposure, similar to the human melanopsin (positive control) (*Figure 5A*). By contrast, using corresponding assays for G$\alpha_s$ or G$\alpha_{i/o}$ activation (*Bailes et al., 2012*; *Bailes and Lucas, 2013*), we detected no activation of either G$\alpha_s$ (*Figure 5—figure supplement 1A*) or G$\alpha_{i/o}$ (*Figure 5—figure supplement 1B*) by *P. dumerilii* r-Opsin1. This indicates that *Platynereis* r-Opsin1 specifically activates G$\alpha_q$, similar to *Drosophila* r-opsins. The relative responsiveness of a photoreceptor cell to different wavelengths of light is a fundamental determinant of its sensory capabilities. We therefore next determined the spectral sensitivity of *P. dumerilii* r-Opsin1 using our HEK293 cells second messenger assay to measure changes in calcium concentration in response to near monochromatic stimuli spanning the visible spectrum (*Figure 5—figure supplement 1C–F*). The EC$_{50}$ values (irradiance required to elicit 50% response; see *Figure 5—figure supplement 1F*) of sigmoidal dose–response curves were converted to a relative sensitivity and fitted with an opsin: retinaldehyde pigment template function (*Govardovskii et al., 2000*). The optimal λ$_{max}$ for the template was determined by least squares as 471 nm (*Figure 5B, C*). *P. dumerilii* r-Opsin1 therefore maximally absorbs light in the blue range, similar to other r-opsin orthologs, such as human melanopsin, which exhibits a λ$_{max}$ of around 480 nm (*Bailes and Lucas, 2013*).

In summary, the presence of all components of the r-opsin phototransduction pathway in TRE cells, and our demonstration that *P. dumerilii* r-Opsin1 is capable of activating G$\alpha_q$, strongly suggests that EP and TRE cells can respond to light.

## Mutation of *r-opsin1* affects TRE-specific, light-dependent expression of an Atp2b calcium channel involved in hearing

In order to gain insight into the function of r-Opsin1 in TRE cells, we generated two independent *r-opsin1* alleles (*r-opsin1$^{\Delta1}$* and *r-opsin$^{\Delta17}$*) in the background of the *pMos{rops::egfp}$^{vbci2}$* strain by TAL Effector Nucleases (TALENs) (*Bannister et al., 2014*), resulting in premature stop codons in the 5′ coding region of the *r-opsin1* gene (*Figure 5D, E*). Founders were outcrossed to wild-type worms (PIN and VIO strains). Subsequently, trans-heterozygous individuals (*r-opsin1$^{\Delta1/\Delta17}$; pMos{rops:: egfp}$^{vbci2\ +}$*) were used to systematically analyze the molecular profile of EGFP-positive head and

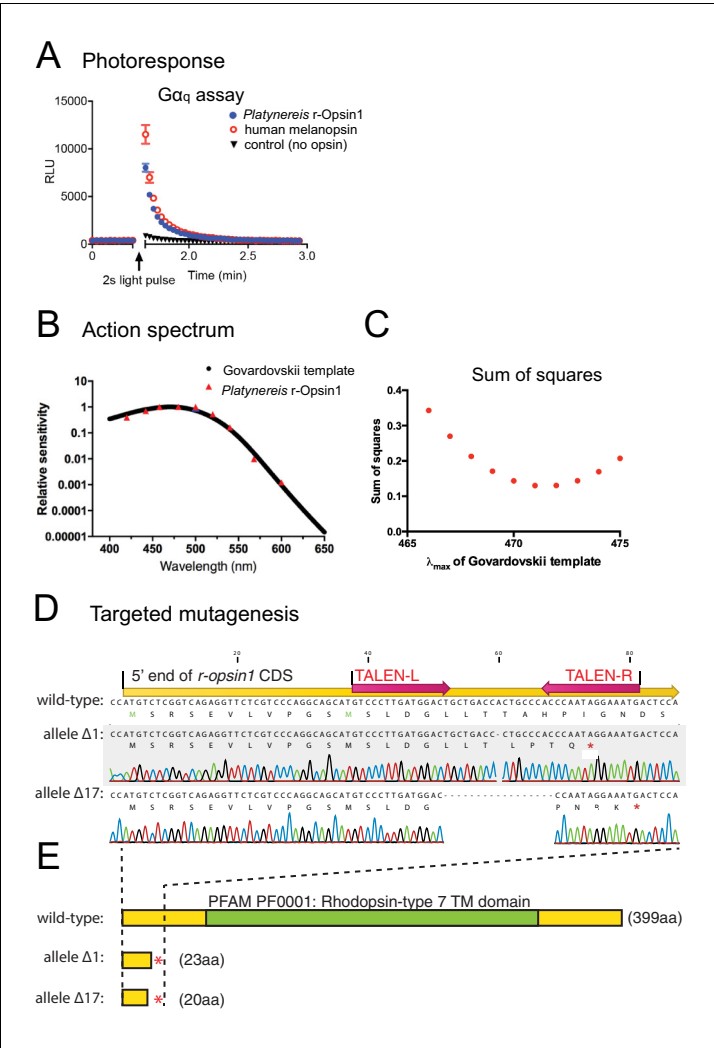

**Figure 5.** Action spectrum and targeted deletion of *Platynereis* r-Opsin1, a Gα_q-coupled blue light photosensor. (A) Gα_q bioassay, showing an increase in luminescent reporter signal for calcium increase after 2 s white light exposure in cells transfected with *Platynereis r-opsin1*. The increase in luminescent reporter signal is similar as when cells are transfected with the positive control human melanopsin. n = 3 independent experiments in all cases. For Gα_s and Gα_{i/o} assays, see **Figure 5—figure supplement 1A, B**. (B) Action spectrum of r-Opsin (based on light spectra and irradiance response curves shown in **Figure 5—figure supplement 1C–F**), fit with a Govardovskii curve visual template obtained with a λ_max of 471 nm. (C) Plotted sum of squares between action spectra and Govardovskii templates at varying λ_max, revealing a minimum for λ_max of 471 nm. (D) Targeted mutagenesis of *Platynereis r-opsin1*. Nucleotide alignment between the 5′ ends of the wild-type (top) and mutant alleles for *r-opsin1*. In the wild-type sequence, positions of the coding sequence (yellow), and of the TALE nuclease binding sites (red arrows) are indicated. Allele Δ1 contains a single-nucleotide deletion, allele Δ17 lacks 17 nucleotides; both lead to premature stop codons (marked as red asterisks in the corresponding translations). (E) A comparison of the encoded proteins (protein lengths indicated in brackets) reveals that alleles Δ1 and Δ17 lack the complete 7-transmembrane domain (green, PFAM domain PF0001) including the critical lysine residue for retinal binding, strongly predicting the alleles as null alleles.

The online version of this article includes the following figure supplement(s) for figure 5:

**Figure supplement 1.** Signaling properties of *Platynereis* r-Opsin1.

---

trunk cells as described above. Sampling from related EGFP-positive non-mutant specimens (*r-opsin1*^{+/+}; pMos{rops::egfp}^{vbci2 +}) served as controls to match mutant vs. non-mutant profiles. Based on our spectral sensitivity results for r-Opsin1, specimens were kept under monochromatic blue light (~470 nm, i.e., the λ_max of r-Opsin1) for 3–5 days until dissociation for FACS.

We next identified the genes differentially expressed between the EP or TRE cells of mutant vs. non-mutant worms using the EdgeR algorithm. Genes with an FDR < 0.05 were considered significantly differentially expressed. We then focused on the *P. dumerilii* homologs of all mouse hearing genes that were expressed in either EP or TRE cells of mutant or non-mutant worms. In the EP cells, none of these candidate genes was significantly differentially expressed between mutant and non-mutant worms. By contrast, one gene (*atp2b/c7424*, *P. dumerilii* homolog of mouse *atp2b2*; *Figure 6—figure supplement 1A*, red arrowhead) was significantly depleted in mutant TRE cells compared to wild-type cells (FDR = 0.010; *Figure 6A*). The specificity of this regulation is further supported by the fact that none of the identified phototransduction components were changed.

The *atp2b2* gene encodes a plasma membrane calcium-transporting ATPase, which is expressed in the stereocilia of mechanosensory cells of the murine cochlea and vestibular system (*Street et al., 1998*). Homozygous *atp2b2* mutant mice show balance deficits and are deaf, while heterozygous mutants show partial loss of auditory ability (*Kozel et al., 2002*). These differential effects caused by different genetic dosages of *atp2b2* are consistent with the possibility that regulation of *atp2b2*

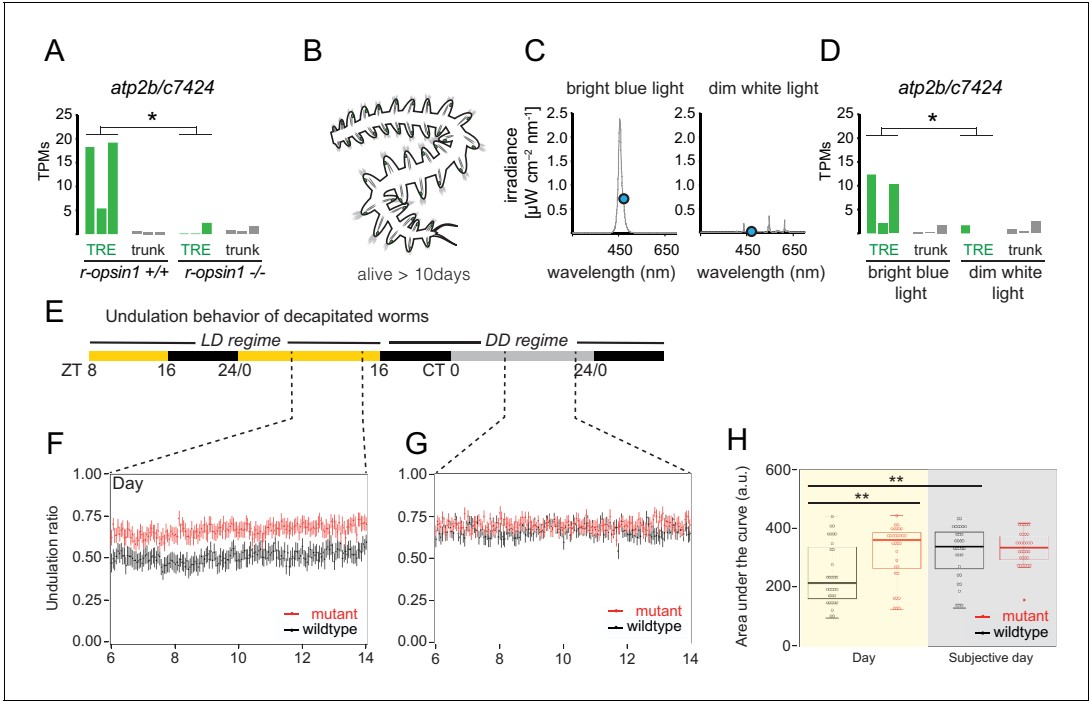

**Figure 6.** *r-opsin1* mediates blue light modulation of trunk *r-opsin1*-expressing (TRE) signature and undulation behavior. (A) *atp2b/c7424* expression levels (in transcripts per million reads [TPMs]) in individual replicates of *r-opsin1+/+* and *r-opsin1-/-* worms cultured for 3–5 days in bright blue light. For Atp2b2 phylogeny, see *Figure 6—figure supplement 1*. (B) Scheme of decapitated worm trunks as used in experiments (D–H) that survive for up to 14 days. (C) Spectral profile of bright blue and dim white light. The blue dot indicates the irradiance at 471 nm ($\lambda_{max}$ of *P. dumerilii* r-Opsin1). (D) *atp2b/c7424* expression levels (in TPMs) in individual replicates of decapitated worms cultured for 3–5 days in bright blue light or dim white light. (E–H) Undulation behavior of decapitated worms. (E) Light regime. Black portions of the horizontal bar indicate 'night' (light off), yellow portions indicate 'day' (light on), and gray portions indicate 'subjective day' (light off during 'day' period). ZT: zeitgeber time; CT: circadian time. (F, G) Undulation ratio during 'day' (F) and 'subjective day' (G). Each black (red) point represents the mean of all wild-type (mutant) worms within a 3 min window, and vertical bars represent the standard error of the mean (n = 32 for each genotype, distributed among three independent experiments). For reliability tests of the algorithm used to detect undulation behavior, see *Figure 6—figure supplement 3*. (H) Area under the curve obtained from the undulation ratios shown in (F) (yellow background; 'Day') and (G) (gray background; 'Subjective day'). Circles indicate data corresponding to individual worms. Boxplots indicate the median (thick horizontal line), the 50% quantile (box), and 100% quantile (error bars). Filled circle indicates an outlier (as determined by the boxplot function of the ggplot R package). *p-value<0.05; **p-value<0.01 (Wilcoxon rank-sum and signed-rank tests). For behavioral responses to strong light of *r-opsin+/+* and *r-opsin-/-* trunks, see *Figure 6—figure supplement 2*.

The online version of this article includes the following figure supplement(s) for figure 6:

**Figure supplement 1.** Atp2b2 phylogeny and enrichment of *atp2b2* in zebrafish neuromasts.

**Figure supplement 2.** Net avoidance crawling distance of decapitated *r-opsin+/+* and *r-opsin-/-* worms in response to strong light.

**Figure supplement 3.** Benchmarking the algorithm used to detect undulation behavior.

expression could be a natural mechanism to modulate mechanosensory cell function. In support of an ancestral role of the plasma membrane calcium-transporting ATPase gene family in modulating neuronal sensitivity, the single *Drosophila* representative of this family, *pmca* (*Figure 6—figure supplement 1A*, violet arrowhead), modulates the thermal sensitivity of motor neurons (*Klose et al., 2009*). Its *C. elegans* ortholog, *mca-3* (*Figure 6—figure supplement 1A*, green arrowhead), modulates touch sensitivity of the touch neurons (*Chen et al., 2015*). Furthermore, in zebrafish, *atp2b2* (*Figure 6—figure supplement 1A*, dark blue arrowhead) is highly enriched in the *r-opsin*-expressing mechanosensory neuromasts of the lateral line (*Figure 6—figure supplement 1B*), consistent with a potential mechanosensory function of the gene in this organism.

Since the expression levels of *atp2b/c7424* in *Platynereis* TRE cells depend on *r-opsin1* function, we wondered if they might also depend on illumination. Decapitated worm trunks are functionally relatively autonomous, maintain their ability to crawl and swim, aspects of their rhythmicity, and live for up to 2 weeks (*Figure 6B*; *Arboleda et al., 2019*). We cultured decapitated trunks of pMos {rops::egfp}$^{vbci2}$ worms for 3–5 days in two distinct light conditions: (i) bright monochromatic blue light (~470 nm) and (ii) very dim white light (*Figure 6C*). Light intensity at the $\lambda_{max}$ of r-Opsin1 (471 nm) was ~40-fold reduced in the dim light condition compared to bright blue light (blue circles in *Figure 6C*). TRE cells were isolated and profiled as before. Statistical analysis showed that *atp2b/c7424* is significantly downregulated in dim light conditions as compared to bright light (p=0.02; Wilcoxon rank-sum test; *Figure 6D*), similar to the downregulation observed in mutant worms (p=0.02; Wilcoxon rank-sum test; *Figure 6A*). These results indicate that blue light levels modulate *atp2b/c7424* expression levels in TRE cells and suggest that the light-dependent modulation is majorly mediated by r-Opsin1.

## r-Opsin1 mediates a light-dependent modulation of undulation frequency

Given the functional relevance of *atp2b2* gene dosage in mammalian hearing, and its enrichment in zebrafish mechanosensory cells known to express *r-opsin* orthologs *opn4xb* and *opn4.1* (*Backfisch et al., 2013*; *Figure 6—figure supplement 1B*), we hypothesized that the regulation of *atp2b/c7424* in TRE cells might correlate with altered mechanosensory abilities. We therefore set out to test the impact of changed light conditions as well as different genotypes on worm behavior. Classical studies have provided evidence for the existence of several classes of mechanosensory cells in parapodia of annelids. These include stretch-sensitive flap receptors, bristle receptors, and acicular receptors (*Dorsett, 1964*; *Horridge, 1963*). A plausible function of these receptors is to fine-tune motor patterns associated with directional (crawling) or stationary (undulation) movements that require coordinated activity by individual segments.

In a first experiment to assess the possible requirement of *r-opsin1* for coordinated segmental movements, we assessed the crawling movement exhibited by decapitated trunks when stimulated by a focal bright light stimulus (*Backfisch et al., 2013*). Trans-heterozygous *r-opsin1*$^{\Delta1/\Delta17}$ individuals clearly responded to such stimuli, but exhibited a significantly reduced net distance when compared to wild-type animals (p=0.02; Wilcoxon rank-sum test; *Figure 6—figure supplement 2*). Whereas this result is consistent with the notion that r-Opsin1 is involved in the correct execution of motor movements, the experiment does not discriminate between r-Opsin1 triggering the response and/or modulating its motor execution.

We therefore decided to investigate a very regularly performed behavior that does not require light as a stimulus. Annelids from the *Platynereis* genus exhibit a stereotypical undulatory behavior that is thought to increase water flow and oxygenation (*Schneider et al., 1992*). The presence of this behavior in *P. dumerilii* is seemingly independent of time (*Zantke et al., 2013*) and requires a tight coordination between segments. Thus, we reasoned that if r-Opsin1 in the segmentally arranged TRE cells plays a role in the modulation of motor movements, this behavior presents a good test. We recorded the movement of *r-opsin1* mutant and wild-type trunks of de-capitated worms for five consecutive days using a previously established infrared video system (*Veedin Rajan et al., 2021*). Concerning visible light conditions, during the first 1.5 days, recorded worms were kept under a light/dark (LD) regime of 16:8 hr, followed by constant darkness (DD, *Figure 6E*). We then established a deep-learning-based quantitative behavioral approach to analyze the resulting movies. We trained a neural network to detect seven different body positions: *jaws, body1-body5, tail* (*Figure 6—figure supplement 3A*) across the total length of each movie. Next, we analyzed 10 s

intervals of the movie to identify oscillatory behavior of the *body1* through *body5* points, using a periodogram algorithm, categorizing each interval into undulatory or non-undulatory behavior. This automated analytical setup was benchmarked against human observations of a portion of the movies (*Figure 6—figure supplement 3B, C*). It allowed us to systematically determine the ratio of time that specimens spent undulating compared to the overall time (*Figure 6F, G*). In turn, this permitted us to compare both the effect of *r-opsin1* mutation (red graphs in *Figure 6F, G*) to wild-types (black graphs in *Figure 6F, G*) and the effect of illumination (day, *Figure 6F*) compared to darkness (subjective day, *Figure 6G*) in equivalent windows of circadian time (CT).

Analyses on a total of 64 trunks revealed that wild-type (black graphs) exhibited a light-dependent modulation of the undulatory movements, which were higher during darkness (*Figure 6F–H*). This modulation was abolished in *r-ops1-/-* worms, whose trunks exhibited equally high undulatory movements during light and dark (*Figure 6F–H*, red graphs). (We noted that in complete animals the difference between wild-type and mutants is also present, but the effect of light modulation on wild-type movements is inversed; data not shown.)

These results hint that in TRE cells r-Opsin1 mediates light-dependent modulation of mechanosensory functions, thereby impacting on regular behavioral movements.

## Discussion

Molecular signatures, like those we derived from the unbiased FAC-sorted EP and TRE cells, as well as from targeted expression analyses, provide valuable information on cell-type divergence and evolution (*Arendt et al., 2019*; *Arendt et al., 2016*; *Liang et al., 2015*). Here, we uncover a serially repeated cell type in the bristleworm trunk that combines r-opsin-dependent photoreception with mechanoreceptive properties. The discovery of this likely multimodal cell type not only represents an interesting example of cellular signal integration, but – as outlined below – also provides fresh thoughts for possible concepts on photo- and mechanosensory cell-type evolution in animals.

Whereas small evolutionary steps can occasionally be experimentally validated, reconstructing evolution across large time scales – as relevant for inferring cell-type evolution since the last common ancestor of vertebrates, flies, and polychaetes – can only be argued based on parsimony. However, which specific scenario in any given case is more parsimonious can be debatable, even among evolutionary biologists.

We thus acknowledge that, even though the statistics of our homology assessments clearly demonstrate that both the photo- and mechanosensory signatures of TRE cells are unlikely to be caused by chance, it remains possible that such a coupling occurred only later in evolution, by co-option of one of these sensory modalities in a cell of the other modality. In such a scenario, the apparent molecular parallels between bristleworm TRE cells, fruitfly JO cells, and zebrafish lateral line neuromasts would then reflect convergent evolution in different branches of animals.

On the other hand, the idea that early animals already possessed complex multifunctional cell types, which later diversified into distinct cells with more restricted and specialized functions, is not unusual for cell-type evolution. Cell types that might reflect such complex ancient states include, for example, the central spinal fluid-contacting neurons of vertebrates that combine sensory processes and hormone/neuropeptide secretion, thereby providing a model for a minimal unit from which multicellular sensory-neurosecretory circuits might have evolved (reviewed in *Arendt, 2008*). Likewise, the eye cells of cnidarian planula larvae have been pointed out as a model of how pigment shading and light reception could have co-existed in a single cell, before cell-type diversification separated photoreceptors from distinct pigment cells (*Arendt, 2008*; *Nordström et al., 2003*). An implicit argument in this view of functional segregation of cell types is that it is easier for evolution to take complex cellular functions apart, or eliminate subfunctions by genetic loss, than to generate complex cell types anew. Likewise, where such functional segregation was accompanied/enabled by the duplication of genes, the resulting homologs may well reflect diverse and specialized functions, while non-duplicated correlates of these genes would be more multifunctional.

In the case of *Platynereis* TRE cells, one plausible explanation for the co-occurrence of mechano- and photosensory features is that such multi-sensory cell type arose early in bilaterian animal evolution. This would predict that molecular similarities are likely to still exist between photoreceptive and mechanoreceptive cells in animals, even if those cell types are nowadays typically distinct.

Indeed, such similarities were previously already noted for shared specification factors such as *atonal/Atonal2/5* and *pou4f3* (*Fritzsch et al., 2007*; *Fritzsch, 2005*; *Piatigorsky and Kozmik, 2004*). Moreover, in line with a model of secondary cell-type diversification, several of the involved distinct regulators arose by gene duplication, such the Pax genes *pax6/ey* – primarily associated with EPs– and *pax2/5/8/spa* – primarily associated with mechanoreceptive cells (*Fritzsch, 2005*; *Niwa et al., 2004*; *Piatigorsky and Kozmik, 2004*), albeit this pattern is not fully consistent across bilaterian phylogeny (*Dobiášovská, 2016*; *Schlosser, 2018*).

In accordance with this notion, development of the worm's TRE cells, found here to exhibit a mechanosensory signature, has been linked to *brn3/pou4f3* and *pax2/5/8* (*Backfisch et al., 2013*). Similarly, the zebrafish lateral line neuromasts that we show to express the *r-opsin1* ortholog *opn4xb* and the *atp2b* ortholog *atp2b2* express *pou4f3* (*Xiao et al., 2005*) and derive from placodes specified by Pax2/8 transcription factors (reviewed in *Schlosser, 2010*). Further, cnidarian PaxB, a transcription factor that combines features of both Pax6/Ey and Pax2/5/8/Spa, has been shown to be involved in the formation of the rhopalia in the Cubozoan jellyfish *Tripedalia cystophora* (*Piatigorsky and Kozmik, 2004*). The rhopalia are sensory structures that combine photo- and mechanosensory functions.

Our hypothesis of the existence of a photo-mechanosensory cell type that predates the split of deuterostomes and protostomes would not only help to explain the close molecular relationships between mechanosensory cells of the lateral line and ear and photosensory cell types present during vertebrate development (reviewed in *Schlosser, 2018*), but also the uncovered genetic links between ear- and eye defects revealed in human conditions such as the Usher syndrome (*Cosgrove and Zallocchi, 2014*), which, without such an evolutionary context, are rather enigmatic.

In contrast to the rather canonical nature of photoreceptor cascades, different animal mechanosensory cells employ different mechanical transducing molecules (*Fritzsch et al., 2020*). As we demonstrate, orthologs of all four main classes of animal mechanical transducing molecules are present in the worm. Three of these are co-expressed with *r-opsin1* in regenerating TRE cells. While *piezo* gene expression is present in most, if not all, (neuro)ectodermal cells of the trunk, including the TREs, *nompc* and *pkd2.1* expression is restricted to few cells. These results support the idea that TREs can indeed function as mechanoreceptive cells. Why different mechanical transducing molecules are co-expressed with *r-opsin1* is not clear. This might indicate heterogeneity among TRE cells. Alternatively, parallel expression of such genes might represent a transient expression feature during regeneration, possibly indicative of cells in the process of specialization. It should be noted, however, that even a transient expression can be evolutionarily meaningful. An example for this is the transient presence of serotonin in substance P-positive neurons in the developing mouse hypothalamus. Whereas such expression does not persist to adulthood in mammals, hypothalamic serotonergic neurons are well-conserved in lower vertebrates (*Tessmar-Raible, 2007*). Future work will be required to further disentangle the diversity of *Platynereis* TRE cells.

Beyond these aspects of cell-type evolution, our results also help to inform the functional evolution of Opsin proteins. The observation that rhabdomeric opsins appear to serve light-independent structural roles in the fly's mechanosensory cells of the JO and ChO has led to the suggestion that such light-independent, cell-mechanical roles are the ancestral function of animal r-opsins (*Katana et al., 2019*; *Zanini et al., 2018*), which contrasts with our hypothesis. We would argue, however, that r-opsins only constitute one of nine opsin families that already existed at the dawn of bilaterian evolution, and light sensitivity is a common feature of its extant members (*Ramirez et al., 2016*). Thus, the evolutionary hypothesis of an ancestral primary non-light sensory function of one bilaterian subgroup either implies that light sensitivity evolved independently in distinct opsin groups or that r-opsins would have undergone a loss of light sensitivity prior to evolving this feature again. A more plausible explanation is that light sensitivity is an ancient feature of r-opsins, and that the close association of r-opsins and certain mechanosensors reflects an ancestral role of light in such cells.

Indeed, our data are consistent with a concept in which opsins endow mechanoreceptors with the ability to tune their responses in response to environmental light conditions, on at least two levels: a first level are light- and *r-opsin1*-dependent changes in transcript levels of *atp2b/c7424*. As ATP2B2 is an ion transport ATPase, which removes $Ca^{2+}$ from the cytoplasm, different expression levels of this enzyme can impact on the time after which a neuron will return to its resting state. Thereby, changing *atp2b* levels likely modulates signal transduction and/or refractory period of cells, resulting

in overall changes in receptor sensitivity. This model is consistent with both the relevance of *r-opsin1* for tuning the undulatory behavior of trunks to ambient light conditions in the bristleworm and the differential effects of different genetic dosages of *atp2b2* (homozygous vs. heterozygous state) in mice (**Kozel et al., 2002**). While we have not directly assessed the speed by which *atp2b/c7424* transcript levels are modulated, such changes would be expected to take place on the scale of minutes to hours, thus providing a slow adjustment of signaling potential.

A second mechanism by which r-opsins could modulate mechanosensation more acutely is provided by the photomechanical response that was uncovered by the study of *Drosophila* EP function (**Hardie and Franze, 2012**). Specifically, this model proposes that opsin-induced, phospholipase C-mediated PIP$_2$ cleavage results in a fast-propagating change in photoreceptor bilayer curvature that then triggers stretch-sensitive TRP-C channels. Thereby, photon absorption (light reception) is effectively translated into a local stretch signal as it is at the core of various mechanosensory cell types. Given that this mechanism seems to account for a canonical photoreceptive function of r-opsin in EP cells, the conservation of *r-opsin* expression along with the respective signaling machinery suggests that opsin activation in mechanoreceptive cells may well acutely tune the membrane curvature and thus the ability of stretch receptors to be activated.

From an ecological perspective, a light-modulatory function could effectively serve to adjust mechanosensory functions in species exposed to varying light conditions, allowing them to tune mechanoreceptive responses to ambient light. Whereas our functional results are restricted to the bristleworm model, we reason that a modulatory function as proposed here might plausibly also reflect the functionality of an ancestral 'protosensory' cell (**Niwa et al., 2004**), which could subsequently have been subfunctionalized into dedicated light sensory and mechanoreceptive cell types. From this perspective, the absence of apparent light sensitivity in *Drosophila* JO or ChO neurons likely represents secondary evolutionary processes rather than ancestral conditions. Likewise, similar principles might apply to the apparent light-independent functions of r-opsins in chemosensory cells suggested by recent experiments in the fruitfly (**Leung et al., 2020**) as chemosensory cells were also noted to share molecular signature with r-opsin light sensors before (**Fritzsch, 2005**). Furthermore, we note that in specific neurons of the cnidarian *Hydra magnipapillata*, the signaling pathway downstream of a distinct opsin class (Cnidops) has been suggested to modulate the discharge of neighboring cnidocytes, a complex cell type also exhibiting sensory functions (**Plachetzki et al., 2012**; **Plachetzki et al., 2010**). It remains unclear if this link reflects parallel evolution or, alternatively, an even more ancient link between opsins and mechanosensory cells. In either setting, however, this finding strengthens the notion that light modulation of animal mechanosensation is a fundamental principle.

Finally, our study also advances technology establishment for a 'non-conventional model system' at multiples levels. First, the FACS-based protocol for cell-type profiling employed here will be useful in the context of other non-conventional marine model organisms. Second, we extend the use of in situ HCR as a sensitive tool for RNA detection in the bristleworm. Third, we anticipate that the automatic analyses of behavioral types by deep-learning-based software tools will provide new opportunities to identify and quantify behavioral paradigms under different environmental and genetic conditions.

# Materials and methods

## Animal culture and handling

All animal research and husbandry was conducted according to the Austrian and European guidelines for animal research (fish maintenance and care approved under BMWFW-66.006/0012-WF/II/3b/2014, experiments approved under BMWFW-66.006/0003-WF/V/3b/2016, which is cross-checked by Geschäftsstelle der Kommission für Tierversuchsangelegenheiten gemäß § 36 TVG 2012 p. A. Veterinärmedizinische Universität Wien, A-1210 Wien, Veterinärplatz 1, Austria, before being issued by the BMWFW). Zebrafish were kept in a constant recirculating system at 26–28°C in a 16 hr light/8 hr dark cycle. Collected embryos were kept at 28°C until hatching.

*P. dumerilii* were raised and bred in the Max Perutz Labs marine facility according to established procedures (**Hauenschild and Fischer, 1969**). Experimental animals were immature adults fed last 4–6 days prior to the day of the experiment. Remaining food was removed a day after feeding, and

the seawater changed, leaving the worms unperturbed for 3–5 days prior to sampling. All pMos {rops::egfp}$^{vbci2}$ transgenic worms (*Backfisch et al., 2013*) used for transcriptome profiling were screened for strong EGFP fluorescence under a stereo microscope system (Zeiss SteREO Lumar V12) at least 6 days before the experiment. To partially immobilize the worms for the screening, worms were shortly transferred to a dry Petri dish.

## Fluorescence-activated cell sorting

EGFP-positive cells from 1 to 2 worms were isolated by FACS with three biological replicates. For the head and trunk of each replica, a sample of unsorted cells was also isolated as reference.

To FAC-sort EGFP+ cells, 1–2 immature transgenic worms per biological replicate were decapitated under a stereoscopic microscope (Zeiss Stemi 2000; Zeiss, Germany) by using a sterile scalpel (Schreiber Instrumente #22; Schreiber Instrumente GmbH, Germany). Separated heads or trunks were placed on ice for about 2 min in 2 ml seawater immediately before dissociation. Heads were mechanically dissociated through a nylon 70 μm cell-strainer (Falcon, USA) in 600 μl seawater. Trunks were first cut into 3–4 pieces using a sterile scalpel, and then dissociated in the same way, using 3 ml seawater. Cell suspensions were passed four times through 35 μm nylon mesh cell-strainers (5 ml polystyrene round-bottom tube with cell-strainer cap, Art. #352235, Falcon) and placed on ice. Finally, the volume of the single-cell suspensions was adjusted to 600 μl (for heads) or 3 ml (for trunks) with ice-cold seawater. Heads and trunks from 1 to 2 non-transgenic worms were also dissociated as negative controls for the detection of EGFP fluorescence.

Cell suspensions were stained with propidium iodide (PI; Thermo Fisher Scientific, P1304MP) by adding 8 μl of 1.5 mg/ml PI per ml of cell suspension and were kept on ice until FAC-sorted. Stained cell suspensions were analyzed on a FACSAria IIIu FAC Sorter (BD Biosciences). FACS events were first gated to exclude aggregates using the FSC-A and FSC-W channels. To separate real EGFP fluorescence from autofluorescence, we followed a previously established strategy (*Revilla-I-Domingo et al., 2018*), measuring fluorescence elicited by a 488 nm laser using two distinct detectors (see *Figure 1C, D*). One quantified fluorescence in the 515–545 nm range ('FITC' axis in *Figure 1C, D*; *Figure 1—figure supplement 1A, B*), while the other quantified fluorescence in the 600–620 nm range ('PE' axis in *Figure 1C, D*; *Figure 1—figure supplement 1A, B*). Comparison between stained cell suspensions from transgenic (*Figure 1C, D*) and wild-type (*Figure 1—figure supplement 1A, B*) specimens allowed for the definition of the gate containing EGFP+ events (boxes in *Figure 1C, D*).

## Transcriptome profiling of EGFP+ cells

Aliquots of 30–120 FACS events from the EGFP+ gate of transgenic heads or trunks were sorted into wells of a 96-well plate (Hard-Shell Low-Profile Thin-Wall 96-Well skirted PCR plate, Bio-Rad HSP-9631) containing 4 μl of lysis buffer. The lysis buffer consisted of 3.8 μl of 0.2% (vol/vol) Triton X-100 (20 μl Triton X-100 BioXtra, Sigma T9284 in 10 ml nuclease-free H$_2$O) + 0.2 μl RNase Inhibitor (Clontech 2313A). Loading of the plate was carried out under a laminar flow hood to avoid contamination, and according to the recommended procedures for subsequent isolation of RNA and synthesis of cDNA using the Smart-Seq2 technology (*Picelli et al., 2014*). The 96-well plate containing lysis buffer was kept on ice until loaded onto the FACS machine. The 96-well plate was maintained at 4°C during the FACS procedure.

A sample of unsorted cells was also taken from the same cell suspension from which the FAC-sorted cells were isolated. For this, immediately prior to FACS, 0.4 μl of the cell suspension was pipetted into 4 μl lysis buffer onto the same 96-well plate used for the sorted cells. From then on, the lysates with FAC-sorted cells and the lysates with unsorted cells were subjected to the same procedures. Immediately after sorting, the 96-well plate containing the lysates was sealed (AlumaSeal CS Films for cold storage, Sigma-Aldrich Z722642-50EA) and stored at −80°C.

After addition of 2 μl of dNTP mix (10 mM each; Fermentas, R0192), 2 μl of oligo-dT-30VN primer (10 μM; 5′–AAGCAGTGGTATCAACGCAGAGTACT30VN-3′), and ERCC spike-in RNA (Ambion) (1:1,000,000 dilution) to the lysates of FAC-sorted or unsorted cells, mRNA isolation, cDNA synthesis with amplification was performed according to the standard Smart-Seq2 protocol (*Picelli et al., 2014*). Single-end 50-bp read sequencing of the cDNA libraries was performed on an Illumina

HiSeq3000/4000 platform according to the manufacturer's protocol. For all samples, transcriptome profiles for three independent biological replicates were obtained.

The different ratios of *eGFP* and *r-opsin1* in the different BRs might be caused by the fact that the time of the day at which the sampling for the different BRs took place differed by about 8 hr and thus spans multiple diel timepoints.

## Bioinformatic analyses

### Transcriptome assembly

All sequencing reads from head or trunk FAC-sorted and unsorted samples from transgenic worms were used to assemble a de novo *P. dumerilii* transcriptome, using the Trinity Software version 2.0.6 (*Grabherr et al., 2011*). Transcripts were filtered for a minimum length of 250 bp. Also, all transcripts that contained overlapping sequences of 50 bp or longer were grouped into clusters. This ensured that each sequencing read (50 bp) could be unambiguously mapped onto a single cluster. For each cluster, we computed nominal transcript length by concatenating the unique sequences within the cluster.

### Mapping reads to transcriptome

Sequencing reads from each individual sample were mapped onto the de novo transcriptome using the NextGenMap program (*Sedlazeck et al., 2013*). Reads that could be mapped onto more than one transcript within the same cluster were mapped only onto one of the transcripts. The number of reads mapped onto each transcript was counted, and counts onto the transcripts within each cluster were added to obtain the total number of reads per cluster. As different transcripts within each cluster likely reflect polymorphisms and splice variants, we refer to these clusters as 'genes.' Gene contigs corresponding to spiked-in sequences (obtained by BLAST against ERCC92 sequences) were removed to obtain the list of *P. dumerilii* genes.

### Determining gene expression levels

To obtain normalized expression levels for each gene, we computed the number of transcripts per million reads (TPMs) as follows: (i) we assigned a nominal transcript length to each gene by concatenating the longest transcript within the cluster with all the non-overlapping sequences of the rest of the transcripts of the cluster; (ii) for each gene, we normalized the read counts to the associated transcript length (in kilo basepairs); and (iii) for each sample, we normalized to the total million reads in the sample. Genes were considered to be expressed in any given sample if they showed $\geq$ 12 TPMs in at least one biological replicate. This threshold is consistent with our enrichment analysis (see below) since it is approximately the minimum expression level required for a gene to be significantly enriched in our differential expression analysis. A gene was considered to be expressed specifically in EP (or TRE) cells if it was expressed in EP (or TRE) cells, and not in TRE (or EP) cells.

### Differentially expressed genes

To identify differentially expressed genes, we used the EdgeR software package, according to the developer's instructions (*Robinson et al., 2010*; *Robinson and Oshlack, 2010*). For each experiment, we used the raw read counts to first filter out all genes that did not have more than one count per million in at least three samples within the experiment, and to then calculate normalization factors for each sample by comparing all samples of the same experiment. Subsequently, we used the quantile-adjusted conditional maximum likelihood (qCML) method to calculate the common and gene-wise dispersion, and the exact test for the negative binomial distribution to test for differentially expressed genes (*Robinson et al., 2010*; *Robinson and Oshlack, 2010*). Only genes with an FDR $\leq$ 0.05 were considered significantly differentially expressed. Genes were considered significantly enriched in EP (or TRE) cells of the head (or the trunk) if they fulfilled the following two criteria: (i) they were identified as differentially expressed between EP (or TRE) cells of the head (or the trunk) and unsorted cells of both head and trunk; and (ii) their expression in EP (or TRE) cells of the head (or the trunk) was higher than in unsorted cells of the head and the trunk. Genes were considered specifically enriched in the EP (or TRE) cells of the head (or the trunk) if they were enriched in the EP (or TRE) cells of the head (or the trunk), and not enriched in the TRE (or EP) cells of the head

(or the trunk). Two genes (c8629 and c14134) were excluded from further analyses because they represent redundant fragments of the *ropsin1* gene.

## Detection of bona fide homologs of *Drosophila* and mouse genes

To systematically assess putative gene homology relationships between *D. melanogaster* or *Mus musculus* and *P. dumerilii*, we used the *tblastn* algorithm to compare all *D. melanogaster* or *M. musculus* protein sequences in the ENSEMBL database (Drosophila_melanogaster.BDGP6.pep.all.fa, 1 March 2016; Mus_musculus.GRCm38.pep.all.fa, 10 March 2016) to all transcripts in our *P. dumerilii* de novo assembled transcriptome. To each *D. melanogaster* or *M. musculus* gene ID, we assigned the *P. dumerilii* gene with the best *tblastn* hit, with a stringent E value threshold of 1E-20.

## Identification of *P. dumerilii* components of the phototransduction pathway

To identify *P. dumerilii* components of the canonical r-opsin phototransduction pathway, we assigned bona fide *P. dumerilii* homologs to the key components of the *D. melanogaster* phototransduction pathway (*Figure 2B*; R-opsin, Gaq, Gb, Gg, NorpA/PLC, INAC/PKC, Trp, Trpl, INAD, Cam, NINAC/MyoIII; Arrestin2, PIP5K). The corresponding ENSEMBL (Drosophila_melanogaster. BDGP6.pep.all.fa, 10 March 2016) gene symbols are as follows: R-opsin: NinaE/Rh3/Rh4/Rh5/Rh6; Gaq: Galphaq; Gb: Gbeta76C; Gg: Ggamma30A; NorpA/PLC: NorpA; INAC/PKC: InaC; Trp: Trp; Trpl: Trpl; INAD: InaD; CaM: Cam; NINAC/MyoIII: NinaC; Arrestin2: Arr2; PIP5K: PIP5K59B. To each *D. melanogaster* protein, we assigned the best *P. dumerilii tblastn* hit, with an E value threshold of 1e-20, as described above. Two proteins (Gg and InaD) had no *P. dumerilii tblastn* hits that satisfied this stringent threshold. Therefore, to assign *P. dumerilii* homologs to these proteins, we lowered the stringency of the E value threshold to 1e-8. To corroborate that *c33855* is a bona fide homolog of Gg (E value 2e-9), we confirmed that this *P. dumerilii* gene is the best *tblastn* hit of the *M. musculus* Gg counterpart (Gng; E value against *c33855*: 2e-9). Similarly, to corroborate that *c7982* is a homolog of InaD (E value 2e-16), we confirmed that this gene is the best *tblastn* hit of the *M. musculus* InaD counterpart (Mpdz; E value against *c7982*: 2e-81).

## Statistical assessment of subset specificity

To assess whether the number of EP- and/or TRE-expressed/enriched genes overlapping with the *P. dumerilii* homologs of a set of N *D. melanogaster* or *M. musculus* genes was meaningful, we generated $10^4$ sets of N randomly picked *D. melanogaster* or *M. musculus* genes, and performed the same analysis as for our real set of N *D. melanogaster* or *M. musculus* genes. We then determined the frequency with which such randomly generated sets resulted in an overlap that was equal to or higher than that found for our real set.

## Molecular phylogenetic analyses

Relevant proteins were identified from the *P. dumerilii* transcriptome with the *tblastn* algorithm using selected animal homologs as query. Predicted sponge proteins were aligned with their counterparts from other animals using MUSCLE (*Edgar, 2004*), and molecular phylogenetic analyses were performed using the IQTREE software (*Nguyen et al., 2015*).

## Analysis and validation of differentially expressed genes

To validate the results of our differential expression analysis, we selected two common EP-/TRE-enriched genes (*ngbl/c10609* and *tmdc/c2433*) and three genes specifically enriched in TRE cells (*f8a/c6996*, *dmd/c7924,* and *trpA/c7677*). The genes selected cover a wide FDR range in our statistical analysis (*Supplementary file 1*). *ngbl/c10609* and *tmdc/c2433* are among the top enriched genes in both EP and TRE samples (FDR = $7 \times 10^{-3}$), whereas *trpA/c7677* (FDR = 0.038) is close to the significance threshold (*Supplementary file 1*). The low FDR values for *ngbl/c10609* and *tmdc/c2433* reflect the high level of expression of these genes in the EP and TRE samples for all three biological replicates, and the low level of expression in the unsorted samples (*Figure 2—figure supplement 1A, B*). From these data, we expected that *ngbl/c10609* and *tmdc/c2433* would be expressed at low levels (or not expressed at all) in any cell type other than EP and TRE cells. We used the established single- or two-color whole-mount in situ hybridization (WMISH) (*Tessmar-Raible et al., 2005*) with *r-opsin1* as reference. Within the head, *r-opsin1* is prominently expressed in the four adult eyes

(*Backfisch et al., 2013*), which is reproduced in our controls (*Figure 2—figure supplement 2C, D*, detected in red). Of note, a dense pigment cup covers the internal portion of each eye that contains the photosensitive outer segments of the retinal photoreceptors (*Fischer and Brökelmann, 1966*). This pigmented area can be seen as a dark area in the eyes (*Figure 2—figure supplement 2C, D*), which partially shields the *r-opsin1* staining. However, due to the localization of the photoreceptor cell bodies (and those of the support cells) outside the pigment cup, gene expression can be assessed in this apparent circle around the pigment cup (broken white contour in *Figure 2—figure supplement 2D*). In this non-pigmented area of the eyes, the red staining for *r-opsin* was clearly discernible (*Figure 2—figure supplement 2D*). Single-color ISH using a probe against *ngbl/c10609* showed expression of this gene in the EPs as well (*Figure 2—figure supplement 2E, F*, blue staining), confirmed by two-color WMISH (*Figure 2—figure supplement 2G, H*, arrowhead).

The TRE cells in the trunk of the worm are apparent as single, *r-opsin1*-positive cells within each parapodium in the ventral flap of the dorsal parapodial arm (*Backfisch et al., 2013*; *Figure 3D, E* and *Figure 2—figure supplement 3A, D, E*, red staining). When tested on trunk samples, the probe for *ngbl/c10609* revealed a similar expression pattern to *r-opsin1* (*Figure 2—figure supplement 3B*, blue staining). Two-color WMISH confirmed the co-expression (*Figure 2—figure supplement 3C*, purple color, arrowhead).

Similarly, a riboprobe against *tmdc/c2433* revealed specific staining in the EP cells as well as single cells within each parapodium in a position consistent with the TRE cells (*Pende et al., 2020*).

Specific expression of the three selected TRE-specific genes, was also validated using single- and double WMISH. A probe against *trpA/c7677* shows no expression in the eyes (*Figure 2—figure supplement 2I, J*), while *trpA/c7677* is detected in small spots in each parapodium (*Figure 2—figure supplement 3F, G*, blue staining). Two-color WMISH shows that one of the spots in each parapodium overlaps with *ropsin1* expression, limited to only part of the cell (*Figure 3E*). *f8a/c6996* and *dmd/c7924* were also expressed in a single cell in the ventral flap of the dorsal arm of each parapodium, in a position that is consistent with the TRE cell (*Figure 2—figure supplement 3H–K*), while expression in the eyes was undetected (*f8a/c6996*; *Figure 2—figure supplement 2K, L*) or extremely weak (*dmd/c7924*; *Figure 2—figure supplement 2M, N*). Along with our set of control genes, these additional validations yield a total of 10 genes that confirm our enrichment analysis (*Supplementary file 1*). The confirmation of *f8a/c6996*, *dmd/c7924*, and *trpA/c7677*, with relatively low level of expression and moderate enrichment FDR values (*Figure 2—figure supplement 1*, *Supplementary file 1*) particularly strengthens the validity of our analysis. It is worth noting that genes expressed at low levels are more likely affected by stochasticity effects during cDNA synthesis and amplification than their highly abundant counterparts. This provides a likely explanation why *dmd/c7924* and *trpA/c7677* are detected, respectively, in two and one of the three biological replicates in TRE cells (*Figure 2—figure supplement 1D, E*).

## Bioluminescence assays to assess Gαq, Gαs, and Gαi/o coupling

To test whether *P. dumerilii* rOpsin1 can activate Gαq, Gαs, or Gαi/o GPCR signaling upon light exposure, we adapted established cell culture second messenger assays (*Bailes and Lucas, 2013*). For this, the *P. dumerilii r-opsin1* gene was heterologously expressed in HEK293 cells. Co-transfected luminescence reporters assessed either activation of Gαq signaling (pcDNA5/FRT/TO mtAeq; expressing *aequorin* as reporter of intracellular calcium *Inouye et al., 1985*) or activation of Gαs or Gαi/o signaling (pcDNA5/FRT/TO Glo22F). Transfected cells were incubated overnight with the chromophore 9-cis retinal in single wells of a 96-well plate. Cells were then incubated with 10 μM coelenterazine (for Gαq) or 0.1 M luciferin (for Gαs or Gαi/o, respectively) in the dark for 2 hr and were subsequently exposed to a 2 s (for Gαq and Gαi/o) or 30 s (for Gαs) pulse of white light. The white light pulse was generated by an Arc lamp, spectrum in *Figure 5—figure supplement 1C*. Raw luminescence was measured from each single well on a Fluostar Optima plate reader (BMG Labtech, Germany). While the well under recording was exposed to the light pulse, all other wells were protected from light with a black sheet. To assess activation of Gαq signaling, luminescence was measured with a resolution of 0.5 s and cycle of 2 s. To assess activation of Gαs signaling, increase in cyclic adenosine monophosphate (cAMP) levels was assessed by measuring luminescence with a resolution of 1 s and cycle of 30 s. To assess activation of Gαi/o signaling, cells were treated with 2 μM Forskolin (Sigma-Aldrich) prior to the light pulse, and decrease in cAMP levels was assessed by measuring luminescence with a resolution of 1 s and cycle of 30 s. The time between the light exposure

to the well and the recording of raw luminescence measurements was approximately 3 s. Measurements taken during the dark incubation preceding the light pulse were used as baseline. As positive controls, we used established constructs for jellyfish Opsin (Gαs assay; *Bailes et al., 2012*), human Rhodopsin 1 (Gαi/o assay; *Bailes et al., 2012*), and human Opn4 (Gαq assay; *Bailes and Lucas, 2013*).

## Measurement of spectral sensitivity of *P. dumerilii* r-Opsin1

To determine the spectral sensitivity of *Platynereis* r-Opsin1, the aforementioned bioluminescence assay was further refined. Bandpass (420, 442, 458, 480, 500, 520, 540, 568, and 600 nm) and neutral density filters (0–3.5) (*Figure 5—figure supplement 1D, E*) were used to deliver defined irradiance doses of distinct wavelengths in 2 s light pulses. Maximum luminescence levels acquired from three independent replicates were plotted against the respective irradiance doses used. Individual irradiance response curves for a given wavelength were then fitted to a sigmoidal dose–response function (variable slope, minimal asymptote value constrained to the average raw luminescence baseline for each wavelength), allowing to derive $EC_{50}$ values (irradiance required to elicit half-maximal luminescence responses) for each wavelength (*Figure 5—figure supplement 1F*). The relative sensitivity at each wavelength was calculated as described in *Bailes and Lucas, 2013*. Likewise, the fitting of data to the Govardovskii visual templates for each wavelength and the determination of the curve with the best fit to the measured data to determine $\lambda_{max}$ of *Platynereis* r-Opsin1 in cell culture (*Figure 5B, C*) followed established procedures (*Bailes and Lucas, 2013*).

## Enrichment of *atp2b2* mRNA expression in zebrafish neuromasts cells

The $TG(pou4f3:GAP-GFP)^{s356t}$ transgenic zebrafish line was used for this experiment, which expresses membrane-targeted GFP under the control of the *brn3c* promoter/enhancer (*Xiao et al., 2005*). 30 transgenic or non-transgenic larvae at 6–10 days post-fertilization were decapitated under a stereoscopic microscope. Trunks were dissociated by incubating them in 0.5% Trypsin-EDTA 10× (59418 C-100ml Sigma-Aldrich) diluted in PBS for 3–4 min, and shearing through a 1 ml pipette tip for an additional 6 min. Cell preparations were filtered once through a 70 µm cell-strainer (Falcon, USA), and three times through 35 µm nylon mesh cell-strainers (5 ml polystyrene round-bottom tube with cell-strainer cap, Art. #352235, Falcon), and were then placed on ice. Cell suspensions were stained with PI (Thermo Fisher Scientific, P1304MP) by adding 8 µl of 1.5 mg/ml PI per ml of cell suspension, and were kept on ice until FAC-sorted. To isolate GFP$^+$ neuromast cells, cell suspensions were analyzed on a FACSAria IIIu FAC Sorter (BD Biosciences) using the same gating strategy as above for the isolation of EGFP$^+$ cells from *Platynereis*. Non-transgenic cell preparations were used to distinguish EGFP$^+$ cells from autofluorescent cells, and therefore be able to accurately design the EGFP$^+$ gate. EGFP$^+$ cells were directly FAC-sorted into RLT lysis buffer (Qiagen). After collection, lysate was vortex for 30 s and stored at −80℃. A sample of unsorted cell preparation was lysed to be used as unsorted sample.

Total RNA was isolated from EGFP$^+$ FAC-sorted and unsorted cell lysates by using the RNeasy mini kit (Qiagen) according to the manufacturer's guidelines, cDNA was synthesized by using the QuantiTect Reverse Transcription kit (Qiagen) according to the manufacturer's guidelines. To measure gene expression levels of *actb*, *opn4xb,* and *atp2b2*, quantitative PCR (qPCR) was performed on 96-well plates in a StepOne Real-Time PCR System (Applied Biosystems) using SybrGreen chemistry (Thermo Fisher Scientific). The total volume of all qPCR reactions was 20 µl. Measured expression levels were used to calculate enrichments, normalizing to the *actb* levels. Statistical significance of enrichment was tested on the QPCR relative number of cycles at threshold (cycles at threshold for *opn4xb* or *atp2b2* relative to *actb*) in EGFP+ samples compared to unsorted samples. Bartlett's test was used to test for equal variance.

## Behavioral analyses
### Light-induced crawling movement

To assess the light-induced crawling response of immature wild-type and *r-opsin1* mutant trunks, we followed a previously established method (*Backfisch et al., 2013*). For both wild-type and *r-opsin1* mutant genotypes, we used the pMos{rops::egfp}$^{vbci2}$ transgenic background. Animals were screened prior to the assay to ensure similar EGFP fluorescence intensity.

## Undulation behavior analysis

Wild-type and *r-opsin1* mutant genotypes were used in the pMos{rops::egfp}$^{vbci2}$ transgenic background. Worms were kept unfed for 3 days prior to the start of the experiment. On the day of the start of the experiment, worms were decapitated and then placed in individual hemispherical concave wells of a custom-made 25-well clear plate (*Ayers et al., 2018*; *Veedin Rajan et al., 2021*). To obtain trunks, specimens were anesthetized by using a 1:1 solution of seawater and 7.5% MgCl$_2$, placed on a microscope slide under a binocular dissecting microscope, and decapitated using a surgical blade (#22; Schreiber Instrumente GmbH, Germany). To increase the chance that decapitated worms could build tubes, the decapitation plane was chosen anterior to the pharyngeal region.

Video recording of worm behavior over several days was accomplished as described previously (*Arboleda et al., 2019*; *Veedin Rajan et al., 2021*). Prior to recording, worms were incubated for 2–4 hr to allow them to build tubes, which is part of their normal behavior. During the recording, worms were subjected to one complete light-dark cycle (16 hr light/8 hr darkness), followed by 4 days of constant darkness. White light was generated by custom-made LEDs (Marine Breeding Systems, St. Gallen, Switzerland), reaching worms with an intensity of $5.2 \times 10^{14}$ photons/cm$^2$/s. Analyses focused on zeitgeber time (ZT) 6–14 of the LD cycle (LD1) and CT 6–14 of the first DD cycle (DD1). ZT0: start of lights on. Worms that had not built a tube during the first hours of the recording or those that had matured by the end of the experiment were excluded from further analysis.

Undulation analysis was performed using positional data of seven discrete body points (*Figure 6—figure supplement 3A*) obtained via a deep-learning-based key point prediction algorithm. The algorithm/neural network was created via the interface of Loopy, developed by loopbio GmbH (Vienna, Austria, http://loopbio.com). For training the network, points were manually annotated using 2740 individual frames obtained from different recordings with the setup described above. To ensure high diversity of the training set, chosen recordings covered different sizes and shapes of worms as well as different times of the day. The subsequent data analysis was carried out in Python 3.7.9 using the SciPy (1.5.2), pandas (1.1.3), and NumPy (1.19.2) packages (*Harris et al., 2020*; *McKinney, 2010*; *Virtanen et al., 2020*).

The positional data was first checked for sufficient prediction coverage: worms for which any single point was annotated in less than 90% of the frames were excluded from further analysis. For the retained individuals, any missing XY values were inferred linearly from nonmissing data. To identify undulation, power spectral density was estimated on 10 s intervals for the position of each body point excluding the jaw and the tail by means of a periodogram. For every point, the dominant frequency within the given time window was determined. A movement was defined as undulation if any of the five body points showed a total movement of 0.5–10 pixels and had a dominant frequency within a range of 0.5–1.5 Hz. Undulation ratios obtained by manually scoring video segments were used to benchmark the automated algorithm (*Figure 6—figure supplement 3B, C*).

All statistical tests were done using R (version 3.6.1). First, from the undulation ratios the area under the curve was calculated for every replicate and then the datasets were tested for normal distribution (Shapiro–Wilk normality test). To determine if there were differences between the groups, either a paired (light vs. dark) Wilcoxon signed-rank test or an unpaired (wildtype vs. mutant) Wilcoxon rank-sum test was conducted. Results were considered statistically significant with a p-value<0.05.

## Conventional in situ hybridization and imaging

In situ hybridization and dual-color in situ hybridization on whole heads and trunk pieces (5–10 segments) of immature worms were performed according to established methods (*Backfisch et al., 2013*). Specifically, fixation was performed in 4% paraformaldehyde/2× PTW (PBS with 0.1% Tween20) for 2 hr at room temperature, and Proteinase K treatment was performed in 100 µg/ml Proteinase K for 5 min (whole heads) or 3 min (trunk pieces) at room temperature. Whole heads and trunk pieces were mounted on glass slides and imaged on a Zeiss Axio Imager with 10× or 40× oil immersion objectives. Single parapodia were cut out of the trunk pieces, mounted on glass slides, and imaged with 10× or 40× oil immersion objectives. A Zeiss Axiocam MR5 camera was used for documentation of stainings.

## In situ HCR-based detection of mechanical transducing factor homologs in regenerating trunks

Wild-type immature worms were cut at a position corresponding to ~2/3 of the trunk length, leading to removal of the posterior ~1/3. Cut worms were allowed to regenerate under standard culture conditions for 10 days.

The posterior part of regenerated worms was cut, and the cut pieces were fixed and treated with Proteinase K, according to the conventional WMISH protocol (*Tessmar-Raible et al., 2005*), with fixation in 4% paraformaldehyde/2× PTW (PBS with 0.1% Tween20) for 2 hr at room temperature, and Proteinase K treatment in 10 µg/ml Proteinase K for 5 min (*Tessmar-Raible et al., 2005*). Expression of *r-opsin1* and/or the candidate mechanosensory genes *nompc, piezo,* and *pkd2.1* in the fixed trunk regenerates was tested using conventional WMISH (*Tessmar-Raible et al., 2005*) and/or an in situ HCR version 3.0 protocol (*Choi et al., 2018*), taking into account recent adaptions for *P. dumerilii* (*Kuehn et al., 2021*).

Specifically, for the HCR protocol, samples were processed in 1.5 ml tubes. Probe hybridization buffer, probe wash buffer, amplification buffer, and fluorescent HCR hairpins were purchased from Molecular Instruments (Los Angeles, USA). Hairpins associated with the b1 initiator sequence were labeled with Alexa Fluor 546, and the hairpins associated with the b2 initiator sequence were labeled with Alexa Fluor 647. To design probes for HCR, we used custom software (*Kuehn et al., 2021*) to create 20 DNA oligo probe pairs specific to *P. dumerilii r-opsin1* (GenBank accession: AJ316544.1), *nompc* (GenBank accession: MZ647694), *piezo* (GenBank accession: MZ647695), and *pkd2.1* (GenBank accession: MZ647696). The *r-opsin1* probe was designed to be associated with the b1 initiator sequence, while the *nompc, piezo,* and *pkd2.1* probes were designed to be associated with the b2 initiator sequence. For the detection stage, samples were washed in 1 ml of 50% probe hybridization buffer in PTW (PBS with 0.1% Tween20) for 5 min at room temperature, pre-hybridized in 300 µl of probe hybridization buffer for 1 hr at 37°C, and then incubated in 300 µl hybridization buffer containing probe oligos (4 pmol/ml) overnight at 37°C. To remove excess probe, samples were washed 4× with 1 ml probe wash buffer for 15 min at 37°C, and subsequently 2× in 1 ml 5× SSCT (5× SSC with 0.1% Tween20) for 5 min at room temperature. For the amplification stage, samples were pre-incubated with 300 µl of amplification buffer for 30 min, room temperature, and then incubated with 300 µl amplification buffer containing fluorescently labeled hairpins (60 pmol/ml each, snap-cooled as described; *Choi et al., 2018*) overnight in the dark at room temperature. To remove excess hairpins, samples were washed in 1 ml 5× SSCT at room temperature, twice for 5 min, twice for 30 min, and once for 5 min. During the first 30 min wash, samples were counterstained with DAPI (Cat. #40043, Biotium, USA), and for final storage moved into 100% glycerol at 4°C. To rule out signal artifacts due to non-specific fluorescence, control samples were treated exactly like the experimental samples (including staining with fluorescent b1-associated amplifier hairpins, named '*b1-amp*' in *Figure 4*, and/or fluorescent b2-associated amplifier hairpins, named '*b2-amp*' in *Figure 4*), except that no probe oligos were added at the detection stage of the protocol ('negative control'). Samples were imaged in 35 mm glass-bottom dishes (P35G-1.5–20 C, MatTek Corporation, USA) using a confocal microscope (LSM 700, Zeiss, Germany) with a 10× or 25× oil objective. FIJI software was used to overlay images and generate z projections.

## Generation of *r-opsin1* mutant strains

The *r-opsin1* genomic region was amplified to screen putative size polymorphic alleles or single-nucleotide polymorphisms (SNPs) from different *Platynereis* strains (PIN, VIO, and ORA) using the following primer combinations: rops1_F1/R1, rops1_F2/R2, rops1_F3/R3, rops1_F4/R4, and rops1_F5/R5. The target alleles or SNPs were screened as described (*Bannister et al., 2014*). *r-opsin1* TALEN pairs were designed in several non-polymorphic exon regions using the TALE-NT prediction tool (*Doyle et al., 2012*). In silico predictions were performed by using customized design conditions, 15 left/right repeat variable diresidue (RVD) length, 15–25 bp spacer length, G substitute by NN RVD, and presence of exclusive restriction enzyme site around the spacer region. The predicted *r-opsin1* TALENs were constructed in vitro using Golden Gate assembly protocol (Golden Gate TAL Effector Kit 2.0, Addgene #1000000024) (*Cermak et al., 2011*). The final TALEN repeats were cloned to heterodimeric FokI expression plasmids pCS2TAL3-DD for left TALEN array and pCS2TAL3-RR for right TALEN array (*Dahlem et al., 2012*). All cloned TALEN plasmids were

sequence-verified using TAL_F1 and TAL_R2 primers. *r-opsin1* TALEN mRNA for each array were made by linearizing the corresponding plasmid by NotI digestion and transcribed in vitro using mMESSAGE mMACHINE Sp6 kit.

Two TALEN pairs targeting exon 1 of *r-opsin1* were designed and generated using the above in vitro assembly protocol. Both *r-opsin1* TALEN spacer regions were flanked with restriction sites (TAL 1 – Bts1 and TAL 2 – Taa1). Following microinjection of 200 ng/µl *r-opsin1* TALEN mRNA, the *Platynereis* embryos were screened for mutations using incomplete restriction digestion and confirmed by sequencing the undigested band. Several injected embryos were raised and outcrossed to wild-type. The F1 outcrossed worms were screened for mutations with a similar restriction digest procedure. Two deletion and insertion mutations were recovered (17 bp deletion and 1 bp deletion). Mutant worms were raised and crossed for several generations to generate both homozygous incross strains and respective wild-type relatives.

### Light and temperature conditions

*r-opsin1*-mutant pMos{rops::egfp}^vbci2 worms and the corresponding pMos{rops::egfp}^vbci2 control individuals used for transcriptomic analysis were incubated without feeding for 3–5 days before the experiment. Blue light of 470 nm was generated using LEDs. The resulting spectrum and intensity of the light was measured using a SpectriLight ILT950 Spectroradiometer (International Light Technologies, MA, USA) (*Figure 6B*). The temperature (kept between 18.5°C and 20°C) was monitored during the 3–5 days of blue light incubation using a HOBO Pendant Temperature/Light Data Logger (Part #UA-002–64, Onset Computer Corporation, MA, USA).

EGFP transgenic worms used for transcriptomic analysis at distinct light conditions were incubated for 3–5 days in blue light or in dim white light after decapitation. Blue light conditions were as described above. Dim white light conditions were obtained by placing the worms in an area partially protected from light within a room with standard white light illumination. The exact spectrum and intensity of the light (see *Figure 6B*) was determined using the same spectroradiometer as described above. The temperature was monitored with a similar device as described above and was kept within the same range as in the blue light conditions (18.5–20°C).

## Acknowledgements

We thank the members of the Tessmar-Raible and Raible groups for discussions, Aida Ćorić and Dunja Rokvic for experimental help, Christoph Bock for support with RNA amplification, and Andrij Belokurov, Margaryta Borysova, and Netsanet Getachew for help with worm care and genotyping at the Max Perutz Labs aquatic facility.

The research leading to these results has received funding from the European Research Council under the European Community's Seventh Framework Programme (FP7/2007–2013)/ERC Grant Agreement 260304 (FR) and ERC Grant Agreement 337011 (KT-R); the research platforms 'Rhythms of Life' (KT-R, FR, AvH) and 'Single-cell genomics of stem cells' (FR) of the University of Vienna; the Austrian Science Fund (FWF) START award, project Y413 (KT-R); Austrian Science Fund (FWF) projects P28970 (KT-R), P30035, I2972 (FR), and SFB F78 (FR and KT-R). AvH and MS acknowledge financial support from the University of Vienna and the Medical University of Vienna. RR was supported by the Vienna International PostDoctoral Program for Molecular Life Sciences (VIPS). None of the funding bodies was involved in the design of the study, the collection, analysis, and interpretation of data or in writing the manuscript.

## Additional information

### Competing interests

Kristin Tessmar-Raible: Reviewing editor, *eLife*. The other authors declare that no competing interests exist.

## Funding

| Funder | Grant reference number | Author |
| --- | --- | --- |
| European Research Council | ERC Grant Agreement 260304 | Florian Raible |
| European Research Council | ERC Grant Agreement 337011 | Kristin Tessmar-Raible |
| European Research Council | ERC Grant Agreement 819952 | Kristin Tessmar-Raible |
| Universität Wien | Research Platform "Rhythms of Life" | Florian Raible Kristin Tessmar-Raible |
| Universität Wien | Research Platform "Single-cell genomics of stem cells" | Florian Raible |
| Austrian Science Fund | START award/project Y413 | Kristin Tessmar-Raible |
| Austrian Science Fund | P28970 | Kristin Tessmar-Raible |
| Austrian Science Fund | I2972 | Florian Raible |
| Austrian Science Fund | SFB F78 | Florian Raible Kristin Tessmar-Raible |
| Austrian Science Fund | P30035 | Florian Raible |

The funders had no role in study design, data collection and interpretation, or the decision to submit the work for publication.

## Author contributions

Roger Revilla-i-Domingo, Conceptualization, Data curation, Software, Formal analysis, Validation, Investigation, Visualization, Methodology, Writing - original draft; Vinoth Babu Veedin Rajan, Data curation, Formal analysis, Investigation, Visualization, Methodology, Writing - review and editing; Monika Waldherr, Data curation, Software, Formal analysis, Investigation, Visualization, Methodology, Writing - review and editing; Günther Prohaczka, Data curation, Software, Formal analysis, Investigation, Methodology, Writing - review and editing; Hugo Musset, Moritz Smolka, Software, Methodology, Writing - review and editing; Lukas Orel, Resources, Formal analysis, Investigation, Writing - review and editing; Elliot Gerrard, Formal analysis, Investigation, Methodology, Writing - review and editing; Alexander Stockinger, Methodology; Matthias Farlik, Resources, Methodology, Writing - review and editing; Robert J Lucas, Supervision, Investigation, Methodology, Writing - review and editing; Florian Raible, Conceptualization, Data curation, Formal analysis, Supervision, Funding acquisition, Visualization, Writing - original draft, Project administration; Kristin Tessmar-Raible, Conceptualization, Supervision, Funding acquisition, Investigation, Visualization, Writing - original draft, Project administration

## Author ORCIDs

Roger Revilla-i-Domingo  https://orcid.org/0000-0001-7943-5776
Vinoth Babu Veedin Rajan  http://orcid.org/0000-0002-2430-7395
Moritz Smolka  http://orcid.org/0000-0002-8621-600X
Matthias Farlik  http://orcid.org/0000-0003-0698-2992
Robert J Lucas  http://orcid.org/0000-0002-1088-8029
Florian Raible  https://orcid.org/0000-0002-4515-6485
Kristin Tessmar-Raible  https://orcid.org/0000-0002-8038-1741

## Ethics

Animal experimentation: All animal research and husbandry was conducted according to Austrian and European guidelines for animal research (fish maintenance and care approved under: BMWFW-66.006/0012-WF/II/3b/2014), experiments approved under: BMWFW-66.006/0003-WF/V/3b/2016.

## Decision letter and Author response

Decision letter https://doi.org/10.7554/eLife.66144.sa1

Author response https://doi.org/10.7554/eLife.66144.sa2

## Additional files

### Supplementary files

- Supplementary file 1. Synopsis of validated genes identified in the transcriptome profiling.
- Supplementary file 2. Sequence identifiers of *Platynereis* genes analyzed in this study.
- Supplementary file 3. Sequence identifiers of protein sequences used for phylogenetic trees.
- Transparent reporting form

### Data availability

All metadata and source files are available for download from Dryad (doi:10.5061/dryad.m63xsj416). This includes raw data, scripts, and the newly assembled and size-filtered transcriptome, used for quantitative mapping (cf. section on Transcriptome profiling).

The following dataset was generated:

| Author(s) | Year | Dataset title | Dataset URL | Database and Identifier |
| --- | --- | --- | --- | --- |
| Revilla-i-Domingo R, Rajan VBV, Waldherr M, Prohaczka Gn, Musset H, Orel L, Gerrard E, Smolka M, Farlik M, Lucas RJ | 2021 | Metadata for the characterization of Platynereis dumerilii cephalic and non-cephalic sensory cell types | https://doi.org/10.5061/dryad.m63xsj416 | Dryad Digital Repository, 10.5061/dryad.m63xsj416 |

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
