## [Decision Letter]

**Acceptance summary:**

This work that will be of significant interest to sensory and evolutionary biologists: It analyzes the roles that opsin proteins might play beyond photo-transduction and how this can inform our understanding of the evolution of sensory neurons. The work examines the transcriptome of opsin-positive cells in a marine annelid and analyzes their behavior. It suggests that these receptors show signatures of both photo- and mechanosensation. This work adds to a growing and surprising realization that single cell types in many different animals may have multi-sensory functionality and this leads to speculation about the evolution of these cell types.

**Decision letter after peer review:**

Thank you for submitting your very interesting article "Analyses of cephalic and non-cephalic sensory cell types provide insight into joint photo- and mechanoreceptor evolution" for consideration by *eLife*. Your article has been reviewed by 4 peer reviewers, and the evaluation has been overseen by a Reviewing Editor and Marianne Bronner as the Senior Editor. The following individuals involved in review of your submission have agreed to reveal their identity: Todd Oakley (Reviewer #1); Gerhard Schlosser (Reviewer #4).

The reviewers had a very lively discussion with one another, and the Reviewing Editor has drafted this to help you prepare a revised submission.

The reviewers found the paper to be potentially of significant interest. However the discussion among reviewers led to two conclusions that must absolutely be addressed before the paper can be considered for *eLife*:

– The first one might be difficult to address: It concerns your argument that the sensory receptors that you describe are also mechanosensory. The reviewers are not convinced by your arguments that are based on the expression of genes that are found in mechanoreceptors. However, the reviewers argue that most (all?) of these genes are also found in other receptors, and in particular in photoreceptors, for instance Glass. Therefore, you absolutely need to provide functional, anatomical or molecular demonstration that these receptors are mechanoreceptors.

We suggest that one of the following unambiguous arguments must be presented:

1) Cell responds to mechanosensory stimulation (e.g., by ephys or calcium imaging).

2) Cell (and possibly associated cells) exhibit one of the highly characteristic morphologies associated with known mechanosensors.

3) Cell expresses known mechanotransducers like Piezos or NompC.

4) Cell mediates behavioral response to mechanosensory stimulation although this would be weaker and indirect evidence.

– The second is the evolutionary argument that must be seriously dimmed and alternative views must be presented as again, there are other possibilities than the one you present.

If you were able to answer the first point (I am sure that you can revise the text to address the second one) we will be able to consider the paper for *eLife.Reviewer #1 (Recommendations for the authors):*

Recommendations for authors. My main recommendation is to be more balanced in the evolutionary interpretations. The authors do explain in quite a bit of detail the possibility of an ancestral hybrid cell type. But they seem to almost be assuming that, rather than discussing any alternative. I do believe that a strong alternative is multiple co-option events that merge phototransduction and mechanosensation multiple times during bilaterian evolution. In cell type evolution, this has been called "fusion". I do not mean to imply I think the authors are wrong in suggesting ancestral multi-functionality. They may well be correct. However, I suspect that going through all the logic in detail would indicate that a similarly convincing case could be made for separate fusion of sensory modalities in flies, annelids, and fish – and perhaps in a case not mentioned here of r-opsin expressed in presumptive mechanoreceptors of Octopus skin. In any event, even if the authors conclude that multiple co-option/fusion events is less logical; I think the manuscript would be stronger in considering alternatives more explicitly.

*Reviewer #2 (Recommendations for the authors):*

The transcriptomes generated are of likely to be of interest to those working on sensory biology in marine annelids, but of limited interest to the broader community. That is because broader conclusions about the roles of r-Opsins outside the visual system are difficult to draw from the data presented, as the data lack detailed functional, physiological or anatomical analysis of the TREs. Broad conclusions about the properties of a protosensory cell are not particularly informed by the data either, given the limited evolutionary scope of the comparisons made.

In terms of suggestions, it would be suggested that the authors present their data for what they are (transcriptomes of r-opsin expressing cells from the marine worm, evidence that the r-ospin is a photosensor and that light alters undulation of a headless worm) and limit the speculations about function and evolution.

Another suggestion is that the authors need to be more sophisticated in interpreting what the patterns of gene expression they observe might mean. For example, the authors cite the expression of Glass as being indicative of "a mechanosensory signature". But Glass is also a widely studied transcription factor important for the *Drosophila* photosystem, so it makes sense it would be co-expressed with an opsin. GO terms can be useful, but the authors should invest more effect into understanding the functions of the genes they are citing to make their arguments.

*Reviewer #3 (Recommendations for the authors):*

There is really an impressive amount of innovative and careful work in this paper. I do believe that the major conclusions of the paper could be strengthened by a larger-scale reconstruction of phylogenetic trees of gene families that appear in both photoreceptor and mechano-sensory receptors.

1. Throughout the text, opsins are capitalized even in the middle of sentences, where they should be lowercase. Suggest going through the paper to correct this.

2. Line 51: should be "an ancient class of opsin".

3. Line 57: Not sure of the correctness of this statement: that the phototransduction cascade consists of only or "a total of 12 proteins". Comparative phototransduction studies in insects have revealed that some *Drosophila* phototransduction gene family members are *Drosophila*-specific or are replaced by paralogs. Also I don't think that available studies of *Drosophila* rule out other proteins involved in phototransduction. The total number of genes involved in phototransduction (for example chromophore metabolism and transport) is likely to be much larger.

4. Lines 69-70: Language here is a bit awkward. Suggest something along the lines of: "This implies an evolutionary question: to what extent do non-cephalic r-opsin-positive cells share an evolutionary history with cephalic eye photoreceptors or represent independent evolutionary inventions?"

5. Figure 2B: why is r-opsin listed twice? Are there two r-opsins or one that is broken into two contigs? If two then please label them differently.

6. Lines 254-256: "This finding indicates that not only rhodopsin genes are present in JO neurons, as reported [15], but that JO neurons possess a complete r-Opsin phototransduction machinery." This statement is written in a confusing way. First, the JO neurons of *Drosophila* were not independently profiled in the current study; rather, the authors seem to be referring to the microarray results of reference 15. How strong is the evidence that gl and wtrw, are mechano-sensory-specific?

7. Lines 272-273: "Indeed, these homologs are significantly overrepresented in the TRE-specific signature." Please somewhere in the paper provide a list of the exact gene names being referred to here. There are many figures which refer varying numbers of genes, but the identity of these seems especially important to the conclusions.

8. Lines 281-282: It would strengthen the interpretation of the paper if phylogenetic trees for each of the 19 TRE-specific gene homologs that have an effect on hearing in mice be generated and presented as supplementary data. For instance, if a majority of these genes were single-copy between mice and Platyneris then this would suggest that the light-dependent mechanosensory hypothesis is correct. But, if a majority of these genes are members of gene family members who are turning over, then the interpretation of independent recruitment for a more derived function cannot be overruled. The authors seem to acknowledge the strength of this independent line of evidence when they mention orthology of transcription factors involved in producing TRE cells/JO neurons and IEH cells in line 300-308.

9. General comment of Figure 3 and Methods 933-953: GO terms and top blast hits are fine for summary figures but in general, when we are talking about comparisons between animals that are this distantly related (Platyneris and *Drosophila* or Platyneris and Danio) I would like to see phylogenetic trees for the 12 (or more) gene families of interest. This is because over evolutionary time, one gene family member may very well be swopped for another and since the authors are making a case that an ancient light-sensitive function of opsin was tied to mechanosensation, these trees need to be shown. As an example: in Figure 5 figure supplement 1A it appears that there are six *Danio rerio* genes that are homologues of the Platyneris Atp2b2 gene but only the function of DrAtp2b2 is mentioned in the manuscript. Are all of these genes in Danio involved in mechanosensation?

*Reviewer #4 (Recommendations for the authors):*

– 103 ff: I think the wording here unnecessarily conflates sensory functions and cell types. Mechanosensation and photosensation may evolve in cell types (e.g multimodal rsensory cells) that are not dedicated mechano- or photoreceptors. So I recommend to replace "mechanoreceptors" in 108 and 110 with "mechanoreception" and take out the clause in 105/106 referring to photoreceptors as separate cell type.

– Figure 1 E/F: I'm a bit puzzled why EGFP and r-opsin levels are not correlated across replicates since EGFP is supposed to be driven by an r-opsin enhancer (why is there a different EGFP/r-opsin ratio between replicates?). Please discuss.

– 294 and Figure 3 Suppl.2: the figure shows that many EP specific genes ad not only TRE specific genes are also related to sound perception in mouse so it appears that TRE specific genes may not be specifically enriched for these. This should be described and discussed.

– 473 ff (also abstract 45 ff): please clarify the use of ancient and ancestral here and in the entire discussion. A cell type or function may be ancient but this does not allow to conclude that it is ancestral for a certain lineage (such a conclusion requires comparison between lineages). If you use "ancestral" please specify for which lineage. I think you can argue tentatively that light-dependent modulation of mechanoreception may be an ancestral bilaterian function even though functional data are lacking for vertebrates and most ecdysozoans.

– 477 ff.: the discussion of the role of POU4f3 and Atonal is oversimplified here since these are required for the specification of rhabdomeric (but not of ciliary) photoreceptors. Thus rhabdomeric PRs seem to share some of their core regulators with mechanoreceptors, while they share Pax6 dependence with ciliary photoreceptors. This presents an interesting conundrum for the phylogenetic relationship between these cell types (discussed in Schlosser, 2018- your reference 68). I know that you don't have the space to discuss this here in detail but a more careful wording is recommended.

– 490: The term "deep homolog" is awkward (and "deep homology" is a bit of a muddled concept in my view); why not just write "homolog" here – (what you are talking about here is proper homology)?

– 595: typo, replace "derived" by "derivation of"

---

## [Author Response]

The reviewers found the paper to be potentially of significant interest. However the discussion among reviewers led to two conclusions that must absolutely be addressed before the paper can be considered for eLife:– The first one might be difficult to address: It concerns your argument that the sensory receptors that you describe are also mechanosensory. The reviewers are not convinced by your arguments that are based on the expression of genes that are found in mechanoreceptors. However, the reviewers argue that most (all?) of these genes are also found in other receptors, and in particular in photoreceptors, for instance Glass. Therefore, you absolutely need to provide functional, anatomical or molecular demonstration that these receptors are mechanoreceptors.We suggest that one of the following unambiguous arguments must be presented:1) Cell responds to mechanosensory stimulation (e.g., by ephys or calcium imaging).2) Cell (and possibly associated cells) exhibit one of the highly characteristic morphologies associated with known mechanosensors.3) Cell expresses known mechanotransducers like Piezos or NompC.4) Cell mediates behavioral response to mechanosensory stimulation although this would be weaker and indirect evidence.

Upon this request, we screened for the orthologs of all four main types of mechanical transducing factors identified in animals, i.e. NompC, Piezo, TMC1/2/3 and Pkd. Of those, members of the Pkd family had already previously been identified in *Platynereis* and shown to be required for mechanosensation in the *Platynereis* larval head (Bezares-Calderón LA et al., *eLife* 2018;7:e36262 doi: 10.7554/*eLife*.36262), at a stage when *r-opsin1* is not yet expressed in the periphery. In short, one member, *pkd2.1* caught our specific attention, because of its already existing peripheral expression, and we selected it for detailed investigation. We also identified orthologs for all the other mentioned genes. The phylogenetic trees are included as Supplementary Figures to Figure 4.

We next reasoned that if our hypothesis of a shared mechano-photosensory cell type is correct, the transcript levels of any mechanical transducing factor in these cells must be low, as we did not detect it by unbiased sequencing. Such low levels are not uncommon for mechanical transducing molecules (see text). We thus decided to investigate the expression in trunk regenerates and with a technique (in situ hybridization chain reaction (HCR)) that is more sensitive than the conventional whole mount in situ hybridization. As HCR in situ hybridization is relatively new for *Platynereis* (at present only a single bioRxiv manuscript exists, describing the usage of this technique on germ cells), we also performed appropriate controls to validate the specificity of the technique. We only included the essential controls in the manuscript for the sake of streamlining, but performed many more.

The combination of regenerates and HCR in situ hybridization then clearly showed that at least two mechanical transducing factors, *pkd2.1* and *nompc*, are highly specifically co-expressed with *r-opsin1* in TRE cells. We think that this is very strong support for our hypothesis of a combined mechano/photosensory nature of the TRE cells.

– The second is the evolutionary argument that must be seriously dimmed and alternative views must be presented as again, there are other possibilities than the one you present.

We carefully considered our colleagues’ comments on this aspect and adapted our text accordingly. Besides a less strong phrasing across the manuscript, we now start the discussion specifically with a paragraph that explicitly acknowledges the possibility of convergent evolution. We also addressed additional points raised by the individual reviewers, see below. We think that this now presents a well-balanced presentation of our hypothesis and the discussions in the evo-devo field.

If you were able to answer the first point (I am sure that you can revise the text to address the second one) we will be able to consider the paper for eLife.Reviewer #1 (Recommendations for the authors):My main recommendation is to be more balanced in the evolutionary interpretations. The authors do explain in quite a bit of detail the possibility of an ancestral hybrid cell type. But they seem to almost be assuming that, rather than discussing any alternative. I do believe that a strong alternative is multiple co-option events that merge phototransduction and mechanosensation multiple times during bilaterian evolution. In cell type evolution, this has been called "fusion". I do not mean to imply I think the authors are wrong in suggesting ancestral multi-functionality. They may well be correct. However, I suspect that going through all the logic in detail would indicate that a similarly convincing case could be made for separate fusion of sensory modalities in flies, annelids, and fish – and perhaps in a case not mentioned here of r-opsin expressed in presumptive mechanoreceptors of Octopus skin. In any event, even if the authors conclude that multiple co-option/fusion events is less logical; I think the manuscript would be stronger in considering alternatives more explicitly.

We thank the reviewer for pointing this aspect out to us. In response to the comment, we carefully revised the discussion, and now start it with a paragraph that more explicitly discusses alternatives to the idea of an ancient cell type. This also allows us to refer more explicitly to other cases of early multifunctional cell types. Together, this should orient the reader better in the field of cell type evolution and allow a more balanced view.

Reviewer #2 (Recommendations for the authors):The transcriptomes generated are of likely to be of interest to those working on sensory biology in marine annelids, but of limited interest to the broader community. That is because broader conclusions about the roles of r-Opsins outside the visual system are difficult to draw from the data presented, as the data lack detailed functional, physiological or anatomical analysis of the TREs. Broad conclusions about the properties of a protosensory cell are not particularly informed by the data either, given the limited evolutionary scope of the comparisons made.In terms of suggestions, it would be suggested that the authors present their data for what they are (transcriptomes of r-opsin expressing cells from the marine worm, evidence that the r-ospin is a photosensor and that light alters undulation of a headless worm) and limit the speculations about function and evolution.Another suggestion is that the authors need to be more sophisticated in interpreting what the patterns of gene expression they observe might mean. For example, the authors cite the expression of Glass as being indicative of "a mechanosensory signature". But Glass is also a widely studied transcription factor important for the *Drosophila* photosystem, so it makes sense it would be co-expressed with an opsin. GO terms can be useful, but the authors should invest more effect into understanding the functions of the genes they are citing to make their arguments.

We understand the first comment of the reviewer as an implicit request for a deeper characterization of TREs on other experimental levels, in order to support their proposed function as mechanoreceptors and thereby put the evolutionary discussion on firmer grounds. In the editorial letter, this request was picked up with a specific set of possible experiments that could provide such additional evidence. In compliance with this list, we now added the clear co-expression of *nompc* and *pkd2.1* with *r-opsin1* to the manuscript. Given that *pkd2.1* has been demonstrated to function as a mechanical transducing factor in larval worms, these data present very strong support for our hypothesis of a dual mechano-/photosensory nature of the TRE cells.

As to the insinuation that we did not read well enough about the functions of the cited genes and only regarded GO-term annotations, the reviewer probably misunderstood our use of the GO-term annotations in the study. We are well aware that hardly any gene has a single biological function. Nonetheless, the involvement in mechanotransduction or mechanosensory cell specification is a specific function assigned to a limited set of genes. Therefore, the significant enrichment for this function in genes matching the specific transcriptome of TRE cells must be interpreted as meaningful, even if the respective genes have additional functions. For the very reason, we also performed analogous comparisons using gene sets for different sensory functions, which did not turn up significant matches (see Figure 3 H, I). To put more emphasis on the statistical assessment, we have now mentioned it in the beginning of the revised discussion. Moreover, the identification of mechanical transducing factors specifically expressed in the TREs (see first part of the comment) now provides independent and specific evidence for the mechanosensory functionality of TRE cells, going beyond the GO term analysis.

Reviewer #3 (Recommendations for the authors):There is really an impressive amount of innovative and careful work in this paper. I do believe that the major conclusions of the paper could be strengthened by a larger-scale reconstruction of phylogenetic trees of gene families that appear in both photoreceptor and mechano-sensory receptors.1. Throughout the text, opsins are capitalized even in the middle of sentences, where they should be lowercase. Suggest going through the paper to correct this.

We thank the reviewer for pointing out the impression of inconsistency. We intended to continue a tradition of naming bristleworm genes and gene products that has been adapted from the zebrafish field: italics would be used for genes and transcripts, capitalized words would refer to specific proteins, and non-capitalized, non-italic words to refer to general groups of proteins. In response to the reviewer’s comments, we realized that we had multiple instances where “opsin” was indeed capitalized while referring to the general class of molecules. We corrected these cases accordingly.

2. Line 51: should be "an ancient class of opsin"

Corrected.

3. Line 57: Not sure of the correctness of this statement: that the phototransduction cascade consists of only or "a total of 12 proteins". Comparative phototransduction studies in insects have revealed that some *Drosophila* phototransduction gene family members are *Drosophila*-specific or are replaced by paralogs. Also I don't think that available studies of *Drosophila* rule out other proteins involved in phototransduction. The total number of genes involved in phototransduction (for example chromophore metabolism and transport) is likely to be much larger.

We thank the reviewer for pointing out that this phrasing was indeed not well done. In fact we meant to refer to “key molecules/components of the phototransduction cascade”, as focused on in the cited reference (Hardie RC, Juusola M. Phototransduction in *Drosophila*. Curr Opin Neurobiol 2015;34:37– 45. doi:10.1016/j.conb.2015.01.008). While this was already correctly phrased in the figure legend of Figure 2, we now also used the more correct phrasing in the manuscript text.

4. Lines 69-70: Language here is a bit awkward. Suggest something along the lines of: "This implies an evolutionary question: to what extent do non-cephalic r-opsin-positive cells share an evolutionary history with cephalic eye photoreceptors or represent independent evolutionary inventions?"

Many thanks. We implemented the reviewer’s suggestion.

5. Figure 2B: why is r-opsin listed twice? Are there two r-opsins or one that is broken into two contigs? If two then please label them differently.

These are indeed two r-Opsins, encoded by distinct genes in *Platynereis*. The distinct IDs were already provided, but we now reformatted the figure to make this clearer.

6. Lines 254-256: "This finding indicates that not only rhodopsin genes are present in JO neurons, as reported [15], but that JO neurons possess a complete r-Opsin phototransduction machinery." This statement is written in a confusing way. First, the JO neurons of *Drosophila* were not independently profiled in the current study; rather, the authors seem to be referring to the microarray results of reference 15.

We rephrased this part.

How strong is the evidence that gl and wtrw, are mechano-sensory-specific?

Besides adding the additional data on putative mechanotransducers in TRE cells, we now also more carefully lay out our reasoning on the different transcripts (including *gl* and *wtrw*) used as indicative for a mechanosensory signature of the TRE cells (see beginning of Discussion section).

7. Lines 272-273: "Indeed, these homologs are significantly overrepresented in the TRE-specific signature." Please somewhere in the paper provide a list of the exact gene names being referred to here. There are many figures which refer varying numbers of genes, but the identity of these seems especially important to the conclusions.

We had already provided the requested lists in Figure 3—figure supplement 1 and in Figure 3—figure supplement 2. (Please note that these files have been re-named to Figure 3- source data 1 and Figure 3- source data 2 in the text). We noticed, however, that we could more clearly refer to these supplementary items in the main text. Thus, we now also refer to them more prominently in the main text, e.g. stating “for gene IDs see Figure 3- source data 2” right after referring to Figure 3G in the sentence that formerly was at lines 272-273.

In case the comment refers to all GO identifiers from *Drosophila* and mouse used for the analysis presented in figure 3- all those had already been up-loaded into the dataset that is available via the DRYAD link at the end of the manuscript.

8. Lines 281-282: It would strengthen the interpretation of the paper if phylogenetic trees for each of the 19 TRE-specific gene homologs that have an effect on hearing in mice be generated and presented as supplementary data. For instance, if a majority of these genes were single-copy between mice and Platyneris then this would suggest that the light-dependent mechanosensory hypothesis is correct. But, if a majority of these genes are members of gene family members who are turning over, then the interpretation of independent recuitment for a more derived function cannot be overruled. The authors seem to acknowledge the strength of this independent line of evidence when they mention orthology of transcription factors involved in producing TRE cells/JO neurons and IEH cells in line 300-308.9. General comment of Figure 3 and Methods 933-953: GO terms and top blast hits are fine for summary figures but in general, when we are talking about comparisons between animals that are this distantly related (Platyneris and *Drosophila* or Platyneris and *Danio*) I would like to see phylogenetic trees for the 12 (or more) gene families of interest. This is because over evolutionary time, one gene family member may very well be swopped for another and since the authors are making a case that an ancient light-sensitive function of opsin was tied to mechanosensation, these trees need to be shown. As an example: in Figure 5 figure supplement 1A it appears that there are six *Danio rerio* genes that are homologues of the Platyneris Atp2b2 gene but only the function of DrAtp2b2 is mentioned in the manuscript. Are all of these genes in Danio involved in mechanosensation?

Both comments refer to the notion that many genes in *Platynereis* (like typical for invertebrates) have multiple orthologs in vertebrates (and sometimes other invertebrates), prompting the question to which extent the function of a specific ortholog can be considered relevant for the single *Platynereis* ortholog.

As to the specific suggestion made in the comment: While we share the reviewer’s interest in detailed phylogenetic trees, we considered the clear prioritization of revision tasks that were presented to us by the reviewing editor, which did not include to generate the about 30 phylogenetic trees requested above. In light of this, we decided to focus on the explicitly requested tasks. Furthermore, there are two aspects we would also ask to consider:

– Phylogenetic trees are included for several key molecules, like for the newly added mechanical transducing molecules.

– One of the evolutionary consequences of vertebrate whole genome duplications are subfunctionalisations among the gene duplicates. Evidences of such subfunctionalisations exist multifold (e.g. pax2/5/8 versus its respective pax2, pax5, pax8 orthologs in vertebrates). In such scenarios, even individual matches of gene functions in one-to-many orthology relationships are evolutionarily meaningful.

Especially on gene level, we consider evolutionary loss (e.g. via the mentioned subfunctionalisation) more parsimonious than de novo gain, as in principle a single mutation is sufficient to destroy the function of a gene, but likely more changes are required to gain a new function.

In any case, we would like to argue that our statistical analyses of enrichment, including tests of other sensory categories, control for biases. Only ‘mechanical stimulation” and “sound” showed statistical significant enrichments for worm versus mouse functionally meaningful orthologs expressed in TRE cells versus mouse mechanoreceptors. If the functional similarities between one-to-many orthologs were just by chance there should not be a higher-than-chance enrichment detectable. We have inserted a brief note on this statistical enrichment at the beginning of the discussion.

Reviewer #4 (Recommendations for the authors):– 103 ff: I think the wording here unnecessarily conflates sensory functions and cell types. Mechanosensation and photosensation may evolve in cell types (e.g multimodal rsensory cells) that are not dedicated mechano- or photoreceptors. So I recommend to replace "mechanoreceptors" in 108 and 110 with "mechanoreception" and take out the clause in 105/106 referring to photoreceptors as separate cell type.

We adjusted the text accord to the reviewer’s request.

– Figure 1 E/F: I'm a bit puzzled why EGFP and r-opsin levels are not correlated across replicates since EGFP is supposed to be driven by an r-opsin enhancer (why is there a different EGFP/r-opsin ratio between replicates?). Please discuss.

This is an interesting question. We think that this is probably due to the fact that the sampling time took app. 8hrs and thus spanned across multiple diel time points. Given evidence from other organisms and also some of our own unpublished observations, it is not unlikely that the levels of *ropsin1* are (moderately?) changing across the day. Depending on the nature of these changes, which can often be post-transcriptional, this might well not be captured by the *r-opsin1::eGFP-f2antr* construct, which does not have the endogenous 3’ UTR of *r-opsin1*. Also, we noticed that the eGFP fluorescence of the construct – while faithfully reproducing the spatial expression- is slightly delayed in regenerating worm tails, compared to the moment when the first *r-opsin1* transcripts become visible by whole mount in situ hybridization. This suggests that while the enhancer is pretty complete, it is slightly delayed in its temporal dynamics. Over the course of the day and in combination with endogenous diel changes of *r-opsin1*, this could result in the observed ratio difference. We now included a sentence on this in the methods’ section.

– 294 and Figure 3 Suppl.2: the figure shows that many EP specific genes ad not only TRE specific genes are also related to sound perception in mouse so it appears that TRE specific genes may not be specifically enriched for these. This should be described and discussed.

This is probably a misunderstanding: while EP specific genes are also present in the GO terms of sound perception and mechanical stimulus, there is no significant enrichment, i.e. this is the number expected to obtain by chance (Figure 3G,J).

– 473 ff (also abstract 45 ff): please clarify the use of ancient and ancestral here and in the entire discussion. A cell type or function may be ancient but this does not allow to conclude that it is ancestral for a certain lineage (such a conclusion requires comparison between lineages). If you use "ancestral" please specify for which lineage. I think you can argue tentatively that light-dependent modulation of mechanoreception may be an ancestral bilaterian function even though functional data are lacking for vertebrates and most ecdysozoans.– 477 ff.: the discussion of the role of POU4f3 and Atonal is oversimplified here since these are required for the specification of rhabdomeric (but not of ciliary) photoreceptors. Thus rhabdomeric PRs seem to share some of their core regulators with mechanoreceptors, while they share Pax6 dependence with ciliary photoreceptors. This presents an interesting conundrum for the phylogenetic relationship between these cell types (discussed in Schlosser, 2018 – your reference 68). I know that you don't have the space to discuss this here in detail but -a more careful wording is recommended.– 490: The term "deep homolog" is awkward (and "deep homology" is a bit of a muddled concept in my view); why not just write "homolog" here – (what you are talking about here is proper homology)?

We thank the reviewer for the valuable comments, which we implemented in the new version of the discussion.

– 595: typo, replace "derived" by "derivation of"

We changed the entire phrase in the figure legends.